# Pathways to school success: Self-regulation and executive function, preschool attendance and early academic achievement of Aboriginal and non-Aboriginal children in Australia's Northern Territory

**Vincent Yaofeng He**[1], **Georgie Nutton**[2]*, **Amy Graham**[2], **Lisa Hirschausen**[3], **Jiunn-Yih Su**[1]

**1** Menzies School of Health Research, Charles Darwin University, Darwin, Northern Territory, Australia, **2** College of Education, Charles Darwin University, Darwin, Northern Territory, Australia, **3** Northern Territory Department of Education, Darwin, Northern Territory, Australia

* georgina.nutton@cdu.edu.au

**Data Availability Statement:** The study datasets contain sensitive personal information and are held

## Abstract

### Background

With the pending implementation of the Closing the Gap 2020 recommendations, there is an urgent need to better understand the contributing factors of, and pathways to positive educational outcomes for both Aboriginal and non-Aboriginal children. This deeper understanding is particularly important in the Northern Territory (NT) of Australia, in which the majority of Aboriginal children lived in remote communities and have language backgrounds other than English (i.e. 75%).

### Methods

This study linked the Australian Early Development Census (AEDC) to the attendance data (i.e. government preschool and primary schools) and Year 3 National Assessment Program for Literacy and Numeracy (NAPLAN). Structural equation modelling was used to investigate the pathway from self-regulation and executive function (SR-EF) at age 5 to early academic achievement (i.e. Year 3 reading/numeracy at age 8) for 3,199 NT children.

### Result

The study confirms the expected importance of SR-EF for all children but suggests the different pathways for Aboriginal and non-Aboriginal children. For non-Aboriginal children, there was a significant indirect effect of SR-EF (β = 0.38, p<0.001) on early academic achievement, mediated by early literacy/numeracy skills (at age 5). For Aboriginal children, there were significant indirect effects of SR-EF (β = 0.19, p<0.001) and preschool attendance (β = 0.20, p<0.001), mediated by early literacy/numeracy skills and early primary school attendance (i.e. Transition Years to Year 2 (age 5–7)).

on a secure cloud-based server with restricted access. Access requires the approval of the ethics committee and data custodians. For applications for data access, please contact the Menzies Data-linkage Program Leader at steve. guthridge@menzies.edu.au.

**Funding:** The study has been supported by an internal research grant by Charles Darwin University College of Education. The funders had no role in study design, data collection and analysis, decision to publish, or preparation of the manuscript.

**Competing interests:** The authors have declared that no competing interests exist.

## Conclusion

This study highlights the need for further investigation and development of culturally, linguistically and contextually responsive programs and policies to support SR-EF skills in the current Australian education context. There is a pressing need to better understand how current policies and programs enhance children and their families' sense of safety and support to nurture these skills. This study also confirms the critical importance of school attendance for improved educational outcomes of Aboriginal children. However, the factors contributing to non-attendance are complex, hence the solutions require multi-sectoral collaboration in place-based design for effective implementation.

## Introduction

Over the past few decades, there has been a growing interest in understanding the pathway to early school success [1]. It is widely acknowledged that early childhood years are a critical time for the development of essential skills that are necessary for successful school transition and subsequent academic achievement. Self-regulation and executive function skills are increasingly regarded as foundational for children's development of social, emotional and cognitive competence to achieve early school success and positive social relationship [2, 3]. These skills are an umbrella term for a range of inter-related components, and complex and simple skills that develop progressively from early childhood. These skills contribute to the ability to manage one's social and emotional experiences [4, 5], also termed 'social and emotional competence' [6]. This study describes the pathways from self-regulation and executive function skills (at age five years) to early school success for children in Year 3 (age eight years) in Northern Territory schools.

### Self-regulation and executive function in school settings

Children who enter school with greater social and emotional competence tend to develop positive attitudes toward school, adjust more successfully to school, attain higher achievement, and be more academically engaged [5]. Teachers are acutely aware of the importance of self-regulation and executive function skills, as the early building blocks for lifelong learning. Often, there is a wide range of capabilities in these skills for children entering school [7]. Earlier research has shown that teachers are most concerned with the self-regulation of school starters, over and above their literacy or academic abilities [8]. Other research has also pointed to social-emotional learning as being a key, foundational area for development within the education system [9]. It is often within the classroom environment, in a group, with the demands of schoolwork that delays or deficits in the development of age-appropriate self-regulation and executive function skills are first noted. Teachers identify that some children may have difficulty with paying attention, managing emotions, completing tasks, and communicating wants and needs verbally, which impacts on their success at school. Even when only one or two students in a classroom have poorly developed self-regulation and executive function skills, the entire class is affected and teacher time is spent on managing behaviour rather than on teaching [7].

**Traits of self-regulation and executive function.**    While related, self-regulation and executive function are distinct, which has been empirically supported [4]. In spite of the established distinction, some authors use the terms interchangeably because of the overlap in core traits

[9]. Both self-regulation and executive function are acknowledged to be multifaceted and spanning across developmental domains. To successfully self-regulate, children switch on and manage a range of skills across cognitive, behavioural, social and emotional domains to achieve desired goals and outcomes. This aspect of self-regulation involves 'effortful control' and the indicators include impulse control, self-control, self-efficacy, responsibility, problem solving, adaptability, and maintaining focus despite the presence of distractions [3, 9, 10]. Self-regulation specifically refers to one's ability to regulate themselves without outside intervention or assistance. Self-regulation also entails as a child's ability to analyse or interpret perceived threats, and respond appropriately, which might include controlling emotional expression [9]. Core concepts of executive functioning include concentration, sequencing and memory, planning; problem solving; delayed gratification and impulse control. Executive function competencies can include: remembering things, like steps in solving a mathematical problem; making plans; focusing on multiple streams of information; making decisions using available information and revising when necessary; resisting urges when they are not appropriate.

**Understanding pathways to school success.**   Due to a broad range of child, family, socio-economic and socio-historical factors that contribute to the growth and consolidation of the essential skills of self-regulation and executive function during early childhood, some children and families need additional support in this process. The critical window for the development of these skills is before children arrive at school, hence major differences in early academic, learning and behavioural regulation skills can emerge at preschool or school entry. A substantial body of research over recent decades has demonstrated the importance of high-quality preschool and early childhood education, particularly for supporting children and families living with disadvantage [11–14].

Early childhood reforms have emphasised the importance of proportional investment in programs to achieve a more civil society. However, despite the extensive literature base that is available on preschool education and its contribution to academic and life outcomes [15], there is currently a dearth of published research on the pathway of self-regulation and executive function skills to academic achievement in middle childhood. In addition, most studies have not accounted for nuanced cultural or contextual differences in pathways into early school achievement. A deeper understanding of the pathways to early school success for different cohorts within a population can inform the targeting of resources to facilitate more equitable early learning outcomes with better effectiveness.

**Unique circumstances in the Northern Territory (NT) of Australia.**   This deeper understanding is particularly important in the Northern Territory (NT) of Australia, where a significant number of children have unique circumstances that require more responsive and differentiated support. The NT's demography is different to other states and territories in Australia, with much greater proportion of children enrolled in remote or very remote schools (48% c/w 3%); with language backgrounds other than English (39% c/w 18%), and Aboriginal or Torres Strait Islander children (hereafter respectfully referred to as Aboriginal children in accordance with the preference of Aboriginal people in the Northern Territory) (40% c/w 5%) at school entry [16, 17]. The demography of NT Aboriginal children is not only different from their non-Aboriginal peers in the NT, but also different from Aboriginal children in other Australian jurisdictions [17]. The majority of Aboriginal children in the NT have language backgrounds other than English and live in remote or very remote regions (i.e. 75–76%), while the majority of non-Aboriginal children in the NT and Aboriginal children in other Australian jurisdictions have English-speaking backgrounds (i.e. 84% and 81% respectively) and do not live in remote or very remote regions (i.e. 76% and 88% respectively) [16, 17]. The significant overlap between Aboriginal background and non-English speaking background in the NT (i.e. 75%) is vastly different from other Australian jurisdictions. Brinkman (2012) found that in

2009, less than 1% of all Australian children (excluding the NT) taking Australian Early Development Census (AEDC) assessment had both Aboriginal heritage and non-English speaking backgrounds [16, 18].

**'Closing the Gap' (on Aboriginal disadvantage) in education.** The socio-historical and political context for First Nations People in Australia has resulted in inter-generational disadvantage and a gap in health, education and economic participation outcomes for many. Ten years after the establishment of targets under 'Closing the Gap' on Indigenous disadvantage in 2008, only 41% of Aboriginal children attended 15 hours of preschool a week in 2018 [19]. Although the proportion of Aboriginal children identified as vulnerable on one or more AEDC domains has decreased from 47.4% in 2009 to 41.3% in 2018, it is still almost twice than that of non-Aboriginal children (i.e. 20.4% in 2018) in Australia [19]. The gap in early developmental outcome between Aboriginal and non-Aboriginal children is the greatest in the NT, with much higher proportion of Aboriginal children identified as vulnerable on one or more AEDC domains (68.3%) than non-Aboriginal children (23.2%) in 2009, with half of these difference explained by potentially modifiable early health and sociodemographic factors [20]. A more recent study found that once these modifiable factors are adjusted for in the multivariable analysis, Indigenous status was not found to be associated with developmental vulnerability, suggesting *"the main influences predicting children's developmental outcome were their experiences of early life health and sociodemographic factors, regardless of their Aboriginal or non-Aboriginal status"* [1]. In 2020, the Australian Government's Closing the Gap annual report showed mixed progress in the targets related to Aboriginal education [21]. While the targets relating to increasing enrolment to early childhood education and Year 12 attainment were on track, the targets relating to school attendance and literacy/numeracy were not met [22]. In the same year, a new National Agreement on Closing the Gap was made [23]. Under the new Agreement, while the original two targets relating to school attendance and literacy/numeracy achievement were excluded, there was a new target relating to increasing the proportion of Aboriginal children assessed as developmentally on track in all five domains of the AEDC to 55% by 2031 [23].

**AEDC: Measure of school readiness at school entry (including social-emotional competence and early academic achievement).** The AEDC is a triennial census of early childhood development at school entry (usually age 5 years) measured across five developmental domains: Physical health and wellbeing, Social competence, Emotional maturity, Language and cognitive skills (school based), and Communication skills and general knowledge [21]. As AEDC measures 'social competence' and 'emotional maturity' for all Australian children in their first year of full-time school (hereafter referred to as Transition in the NT) [24], it has great potential to give the clearest picture of children's social-emotional wellbeing at a population level. It is therefore appropriate to select and investigate 'best fit' indicators of the 'self-regulation and executive function' construct from the AEDC. In fact, AEDC data have been previously used to investigate the link between social-emotional behaviours (i.e. adaptive and maladaptive) at Transition and subsequent academic achievement at Year 3 (i.e. age eight years) [4, 25]. Similarly, it has also been used to measure early literacy/numeracy skills (at age five years) [4, 25]. However, these studies did not explore the specific traits of self-regulation and executive functioning; nor did they include preschool and early years attendance (as covariates) which have been established in a previous study to be critical factors of school success for Aboriginal children in the NT [1].

**Research questions of study.** To address the current research gap, this study aimed to investigate the pathways from self-regulation and executive function (at age five years) to early academic achievement (at age eight years) for both Aboriginal and non-Aboriginal children in the NT. The research questions included:

1. Are self-regulation/executive function and preschool attendance associated with Year 3 reading/numeracy, and to what extent?

2. Do early literacy/numeracy skills and early primary school attendance mediate the association between self-regulation/executive function and Year 3 reading/numeracy, and to what extent?

3. Do the pathways to Year 3 reading/numeracy differ by different demographic characteristics (i.e. sex, non-English speaking background, remoteness, and socio-economic status)?

## Methods

This is a retrospective observational cohort study using linked de-identified administrative datasets in the NT Child Youth and Development Research Partnership (CYDRP) data repository. The data repository and its linkage process has been reported elsewhere [26–28]. Our study cohort consisted of children who had received AEDC assessments (Cycle 1 and Cycle 2 in 2009/10 and 2012 respectively), attended public preschool and school (from first year of formal schooling, the Transition year, to Year 3), and participated in Year 3 National Assessment Program for Literacy and Numeracy (NAPLAN) test in the NT.

The study was approved by the Human Research Ethics Committee of the NT Department of Health and the Menzies School of Health Research (HREC-2016-2708) and Charles Darwin University Human Research Ethics (H19104).

### Data sources

Three administrative datasets were used in this study: a) NT component of the national AEDC dataset; b) NT school attendance dataset, an administrative dataset containing daily attendance records of students enrolled in all NT public schools over the period 2005–2016; c) National Assessment Program for Literacy and Numeracy (NAPLAN) dataset, which contained Years 3, 5, 7 and 9 test results of students in both public and private schools in the NT [29].

### Measures

**Self-regulation and executive functioning.**   The indicators of self-regulation and executive function at Transition were obtained using items from nine sub-domains in the AEDC, with eight sub-domains from the Social Competence and Emotional Maturity domains, and one from the Language and Cognitive Skills domain [30]. These nine sub-domains included "Overall social competence, Responsibility and respect, Approaches to learning, Readiness to explore new things, Pro-social and helping behaviour, Anxious and fearful behaviour, Aggressive behaviour, Hyperactivity and inattentive behaviour, and Interest in literacy/numeracy and memory" (see S1 Table in S1 File). The standardised score for each AEDC sub-domain ranged from 0 to 10, with higher scores indicating better development. In the descriptive analysis (i.e. Table 1 and S2 Table in S1 File), the proportion of children identified as developmentally vulnerable in each of the sub-domains (i.e. scored in the bottom 10% of the national AEDC population) was presented [31]. In the SEM, the standardised score from each of the nine AEDC sub-domains was used as manifest indicator variables for the latent construct 'self-regulation and executive function'.

**Early literacy/numeracy skills.**   Similar to Collie et al. (2019) [4], this study utilised items from three sub-domains of the Language and Cognitive Skills domain (i.e. Basic Literacy, Advanced Literacy and Basic Numeracy of the Language and Cognitive Skills domain) in the

**Table 1. Demographic characteristics of NT children who received AEDC assessments, attended public school (from preschool to Year 3) and participated in Year 3 NAPLAN test.**

| | All (n = 3,199) | Aboriginal (n = 1,432) | non-Aboriginal (n = 1,767) |
|---|---|---|---|
| **Demographic characteristics (%)** | | | |
| Male | 49.5 | 49.2 | 49.8 |
| non-English speaking background | 40.8 | 70.2 | 16.9 |
| Lived in remote/very remote regions | 47.0 | 75.3 | 24.0 |
| Lived in most socio-economic disadvantaged regions | 29.5 | 51.0 | 12.1 |
| **Proportion of children developmentally vulnerable (%)** | | | |
| **Self-regulation and executive function** | | | |
| Overall Social Competence | 9.3 | 13.9 | 5.8 |
| Responsibility and respect | 18.5 | 28.7 | 10.8 |
| Approaches to learning | 12.8 | 20.6 | 6.9 |
| Readiness to explore new things | 11.3 | 14.8 | 8.6 |
| Pro-social and helping behaviour | 13.2 | 19.9 | 8.3 |
| Anxious and fearful behaviour | 12.1 | 15.9 | 9.2 |
| Aggressive behaviour | 18.8 | 30.3 | 10.1 |
| Hyperactive and inattentive behaviour | 16.2 | 23.7 | 10.5 |
| Interest in literacy/numeracy and memory | 11.0 | 17.4 | 6.2 |
| **Early literacy/numeracy skills** | | | |
| Basic literacy | 23.0 | 41.2 | 9.2 |
| Advanced literacy | 19.8 | 32.9 | 9.9 |
| Basic numeracy | 26.8 | 47.2 | 11.3 |
| **Median of school attendance (out of 100)** | | | |
| Preschool attendance | 85.1 | 65.8 | 90.9 |
| Early year attendance | 90.1 | 77.3 | 93.5 |
| **Proportion of children at/above NMS (%)** | | | |
| Year 3 NAPLAN Reading | 72.0 | 48.3 | 91.2 |
| Year 3 NAPLAN Numeracy | 76.2 | 52.0 | 95.8 |

Note: The proportion of children identified as developmentally vulnerable in each of the AEDC sub-domains was calculated based on non-missing records. The proportion of missing data ranged from 3.3%-3.4% for the sub-domains used as the manifest indicator variables for the latent construct 'early literacy/numeracy skills', and ranged from 3.1%-6.3% for the sub-domains used as the manifest indicator variables for the latent construct 'self-regulation and executive function' (refer to S3 Table in S1 File for the proportion of missing data for each of the individual sub-domains). The calculation of proportion of children living in most socio-economic disadvantaged regions exclude one children with missing socio-economic information.

AEDC to measure early literacy/numeracy skills at Transition. (see S1 Table in S1 File). In the descriptive analysis (i.e. Table 1 and S2 Table in S1 File), the proportion of children identified as developmentally vulnerable in each of the sub-domains was presented. In the SEM, the standardised score from each of the three AEDC sub-domains was used as manifest indicator variables for the latent construct 'Early literacy/numeracy skills'.

**Attendance at preschool and early years.** Students' preschool and early years (Transition to Year 2) attendance rates were drawn from the NT school attendance dataset. The attendance rate for each individual child was calculated by dividing the total number of days attended by the total number of expected days of attendance at school.

## Year 3 reading/numeracy

For the descriptive analysis (i.e. Table 1 and S2 Table in S1 File), a binary outcome, at or above National Minimum Standard (NMS), was chosen for interpretability, in which the NMS

represents the benchmark for the basic level of knowledge and understanding that a student requires to function at the specific year level in Australia [29]. Under the NAPLAN assessment scale, there are 10 bands, and the second lowest band reported for each year level represents "the national minimum standard expected of students at that year level", which is "the agreed minimum acceptable standard of knowledge and skills without which a student will have difficulty making sufficient progress at school" [29]. In the SEM, the logit scores were used due to their superior distributional properties by comparison with other types of scores as demonstrated in a previous study [1]. The scores from the 'reading' and 'numeracy' NAPLAN test was used as manifest indicator variables for the latent construct 'Year 3 reading/numeracy'.

## Aboriginal status

Informed by prior research [1, 6] and due to the different demographic characteristics and pathways (to academic outcomes) of Aboriginal and non-Aboriginal children in the NT (as described in the Introduction section), all analyses were stratified by Aboriginal status. In our study, the Aboriginal status variable determined with an algorithmic approach using the same Aboriginal status variable in every dataset of the CYDRP data repository based on their respective demonstrated levels of accuracy, firstly using health datasets, followed by child protection data, and then education and youth justice records [1]. This hierarchy of accuracy was based on systematic evaluation of the completeness and quality of each dataset referenced to health records (i.e. hospital data) for which an audit, in 2011, demonstrated 98% consistency between recorded Aboriginal status and patient interview [32]. The aforementioned approach is described in detail elsewhere [1] and is consistent with best practice guidelines involving data linked from two or more datasets [33].

## Covariates

Covariates used in this study were drawn from the AEDC data, including: sex, language background (English speaking background or non-English speaking background), remoteness (urban or remote, explained below), and socio-economic status. The levels of relative remoteness and socio-economic status were determined according to the Australian Statistical Geography Standard (ASGS) of Australian Bureau Statistics (ABS) at the level of Statistical Area Level 2 (SA2) using children's home address at the time of undertaking the AEDC. The level of remoteness was determined according to the ABS Accessibility and Remoteness Index of Australia (ARIA+)) [34], and the socio-economic status was measured with the Index of Relative Socio-Economic Disadvantage (IRSD), representing socio-economic disadvantage [35].

For analysis that included stratification by socio-economic status, we divided the children into two categories: the most socio-economically disadvantaged regions (the most disadvantaged deciles of IRSD) and other less socio-economically disadvantaged regions (the other nine deciles). In the NT, there are no metropolitan and inner regional areas, the top two of the five categories in ARIA+ [34]. In this study, the outer regional category was re-categorised as "urban" and the remote and very remote regions "remote" [34]. In the NT, the outer regional area comprised the Greater Darwin region [36], while the rest of NT belong to the remote and very remote categories under ARIA+ [34].

## Data analysis

All data analyses were conducted using Stata for Windows, Version 15 [37].

Confirmatory factor analysis (CFA) was performed first to determine the structure of factors and latent correlations among all variables. After the CFA, modification indices were used to identify the sub-domains that might have correlated errors. To understand the different

pathways from self-regulation and executive function to Year 3 reading/numeracy, structural equation model (SEM) was used. In this study, two models were developed: the direct path model and the mediation model.

In the direct path model, we examined the direct effects of self-regulation and executive function and preschool attendance on Year 3 reading/numeracy, after controlling for the covariates and allowing error terms among specific sub-domains to be correlated. Self-regulation and executive function, and preschool attendance were correlated to control for shared variance. In the mediation model, we investigated the mediation effects of early literacy/numeracy skills and early years attendance by examining the direct and indirect effects of self-regulation and executive function and preschool attendance on Year 3 reading/numeracy, in addition to the direct effects of early literacy/numeracy skills and early years attendance on Year 3 reading/numeracy. The decision to have early literacy/numeracy and early years attendance as mediators in the pathway was based on two previous studies [1, 25]. Collie (2018) found that early literacy/numeracy (at age 5) had a mediating role between prosocial behaviour and Year 3 academic achievement (i.e. NAPLAN) [25], while Silburn (2016) found that early years attendance has a mediating role between preschool attendance and Year 3 NAPLAN for NT Aboriginal children [1]. The shared variance between self-regulation and executive function and preschool attendance, and the shared variance between the error terms of the specific sub-domains (similar to direct path model), the early literacy/numeracy skills and early years attendance were correlated in the mediation model to control for the shared variance.

In the SEM, the standardised beta coefficients ($\beta$), rather than unstandardized beta coefficients, were reported. If the standardised beta coefficient ($\beta$) in the pathway from variable A to variable B is 0.5, then for one SD increase in A, B will increase by 0.5 SD. This indicates that when variable A increases by one SD from its mean, variable B can be expected to increase by 0.5 its own SD from its own mean while holding all other relevant variables constant. In reporting unstandardized beta coefficients, when variable A increases by one unit, variable B would be expected to increase by 0.5 unit, while holding all other relevant variables constant. Due to the different scales of the different variables in our study, it is essential to report the standardised beta coefficients to ensure consistent comparison of the path amongst different variables. Standardised beta coefficients ($\beta$) equal to or greater than or equal to 0.10 and 0.25 were considered evidence of moderate and large effect size respectively [38, 39].

In our analysis, there was no missing data for Aboriginal status, sex, language background, remoteness, preschool attendance, early years attendance, Year 3 NAPLAN reading scores and numeracy scores. There was only 1 record missing data for the 'socio-economic status' variable. Missing data for the scores in the three sub-domains (that were used to construct early literacy/numeracy skills latent construct) ranged from 3.3%-3.4% (S3 Table in S1 File). Missing data for the scores in the nine sub-domains (that were used to construct the self-regulation and executive functioning latent construct) ranged from 3.1% to 6.3% (S3 Table in S1 File).

To account for missing data and possible non-normality of the data, we used the robust maximum likelihood with missing values (MLMV) estimator. MLMV produces maximum likelihood estimation accounting for standard errors and the chi-square test statistics for non-normality [40] and allows path-wise complete case analysis, i.e. analysis of all cases with data available for the variables involved in defining each of the paths in the model. To evaluate model fit, we followed Hu and Bentler's [41] recommendations for adequate and good model fit. Root mean square error of approximation (RMSEA) values of less than or equal to 0.08 and 0.05 were considered evidence of adequate and good fit, respectively [41]. Comparative Fit Index (CFI) and Tucker Lewis Index (TLI) values of equal to or greater than or equal to 0.90 and 0.95 were considered evidence of adequate and good fit, respectively [40]. In the multi-group analysis of associations between demographic characteristics to test the difference in

direct effects and indirect effects between different subgroups in the mediation model, the Wald test of linear hypotheses and Wald test of nonlinear hypothesis were run respectively (using the test and testnl commands in Stata) after the model estimation.

## Results

### Descriptive analysis

After applying the selection criteria, the study cohort consisted of 3,199 children having linked records in all included datasets (Fig 1). Of these, 44.5% were Aboriginal, 49.5% were boys, 40.8% had non-English speaking backgrounds, 47.0% lived in remote or very remote regions and 29.5% lived in the most socio-economically disadvantaged regions at the time of their AEDC assessment at around age five years. The socio-demographic factors of geographic remoteness and first language profiles create two dominant and distinct population groups in the NT for the exploration of pathways to academic outcomes. Characteristics of the different groups of children excluded from the study are available in the supplementary material.

In the study cohort, compared to non-Aboriginal children, Aboriginal children were more likely to be speaking English as a second language (70.2% versus 16.9%, $p < 0.001$), live in remote/very remote regions (75.3% versus 24.0%, $p < 0.001$) and reside in the most socio-economically disadvantaged regions (51.1% versus 12.1%, $p < 0.001$). The proportion of boys in the cohort was similar in the Aboriginal (49.2%) and non-Aboriginal cohorts (49.8%, $p = 0.719$).

The group of children who did not participate in either the Year 3 NAPLAN reading and/ or numeracy tests despite having a Year 3 NAPLAN record (n = 846) were most likely to having missing data (ranging from 17.7% to 21.7%) (S3 Table in S1 File) and more likely be

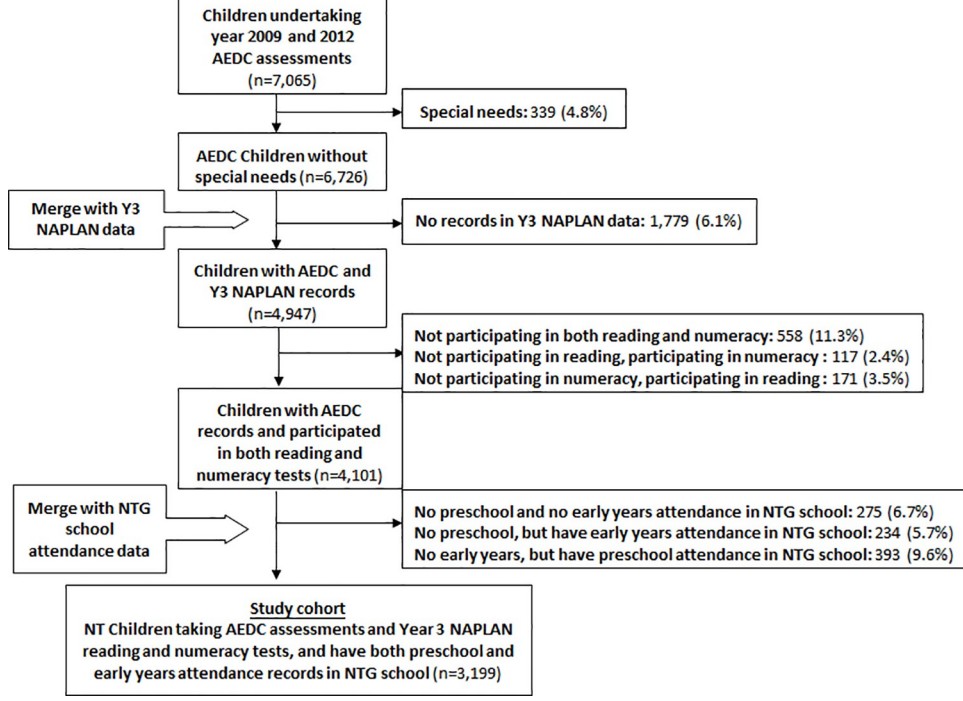

**Fig 1. Flow diagram of cohort selection, NT children receiving AEDC assessments (Cycle 1 and 2 in 2009/10 and 2012).**

developmentally vulnerable (ranging from 19.7% to 59.3%) in all the relevant sub-domains for the latent construct of self-regulation and executive function and early literacy/numeracy skills (S2 Table in S1 File).

Table 1 reports the proportion of children (%) developmental vulnerable on each of the AEDC sub-domains, median school attendance rate (i.e. preschool and early year attendance), and proportion (%) of children at/above NMS (i.e. Year 3 NAPLAN reading and numeracy) for both Aboriginal and non-Aboriginal children in the study cohort. The descriptive results of the Aboriginal and non-Aboriginal children stratified by the demographic variable (i.e. sex, non-English speaking background, remoteness, and social-economic status) were presented in S4 and S5 Tables in S1 File respectively. Good reliability of measures, in both Aboriginal and non-Aboriginal children were demonstrated using Cronbach's alpha for self-regulation and executive function (Aboriginal children $\alpha = 0.89$; and non-Aboriginal children $\alpha = 0.89$); early literacy/numeracy scores (Aboriginal children $\alpha = 0.85$ and non-Aboriginal children $\alpha = 0.80$), and Year 3 reading/numeracy scores (Aboriginal children $\alpha = 0.77$ and non-Aboriginal children $\alpha = 0.80$). S6 Table in S1 File presents the correlations for sex, English speaking background, remoteness and socio-economic status in these two sub-cohorts of children.

## Pathway analysis

**The direct path model.** The CFA in this model yielded a good fit for Aboriginal children ($\chi^2(73) = 411.10$, $p < 0.001$, RMSEA = 0.057, CFI:0.96, TLI:0.94, CD:0.403) and an adequate fit for non-Aboriginal children ($\chi^2(73) = 715.01$, $p < 0.001$, RMSEA = 0.071, CFI:0.94, TLI:0.91 CD:0.089). Self-regulation and executive function had a positive relationship with Year 3 reading/numeracy for both Aboriginal and non-Aboriginal children (Fig 2, Table 2). At the same time, the positive association between preschool attendance and Year 3 reading/numeracy was stronger in Aboriginal children than in non-Aboriginal children.

**The mediation model.** The CFA in this model yielded a good fit for Aboriginal children ($\chi^2(125) = 629.61$, $p < 0.001$, RMSEA = 0.053, CFI:0.96, TLI:0.94, CD:0.432) and an adequate fit for non-Aboriginal children ($\chi^2(125) = 1111.65$, $p < 0.001$, RMSEA = 0.067, CFI:0.93, TLI:0.90, CD:0.13). There are different pathways for Aboriginal and non-Aboriginal children (Fig 3, Table 2). For non-Aboriginal children, the effect of self-regulation and executive function on Year 3 academic outcomes was mainly mediated by early literacy/numeracy skills. For Aboriginal children, both early years attendance and early literacy/numeracy skills appeared to mediate this effect.

For Aboriginal children, there was a significant indirect effect of self-regulation and executive function on Year 3 academic outcomes ($\beta = 0.19$, $p < 0.001$) and an indirect effect of preschool attendance on Year 3 reading/numeracy ($\beta = 0.20$, $p < 0.001$) through early literacy/numeracy skills and early years attendance (Table 2). As shown by the non-significant direct effect of self-regulation and executive function and preschool attendance in the mediation

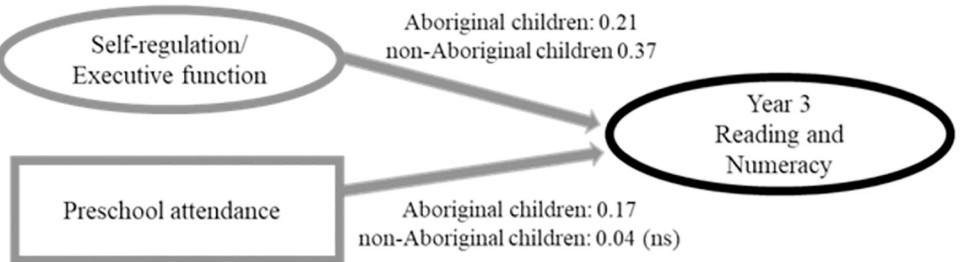

**Fig 2. The direct path model.** All path coefficients are standardised.

**Table 2. Standardised direct and indirect effects (i.e. _β_) of self-regulation/executive function and preschool attendance on Year 3 academic outcomes with early literacy/numeracy skills and early years attendance as mediators.**

| | Aboriginal | non-Aboriginal |
|---|---|---|
| **Direct path model** | | |
| **Direct effects/Total effects** | | |
| Self-regulation and executive function | 0.21 | 0.37 |
| Preschool attendance | 0.17 | 0.04(NS) |
| **Mediation model** | | |
| **Direct effects** | | |
| Self-regulation and executive function | 0.03(NS) | 0.02(NS) |
| Preschool attendance | -0.04(NS) | -0.01(NS) |
| Early literacy/numeracy skills | 0.23 | 0.54 |
| Early years attendance | 0.29 | 0.05(NS) |
| **Indirect effects** | | |
| Self-regulation and executive function | 0.19 | 0.38 |
| Preschool attendance | 0.20 | 0.05(NS) |
| **Total effects** | | |
| Self-regulation and executive function | 0.22 | 0.40 |
| Preschool attendance | 0.16 | 0.04(NS) |

(NS) = Not statistically significant

model, the effect of self-regulation and executive function and preschool attendance on Year 3 reading/numeracy for Aboriginal children was fully mediated by early literacy/numeracy skills and early years attendance.

For non-Aboriginal children, there is a significant indirect effect of self-regulation and executive function on Year 3 reading/numeracy ($β$ = 0.38, $p$ <0.001) through early literacy/numeracy skills. The effect of self-regulation and executive function on Year 3 reading/numeracy was fully mediated by early literacy/numeracy skills (Table 2).

**Association between demographic characteristics and various measures.** For both Aboriginal and non-Aboriginal children, boys scored lower on self-regulation and executive function than girls. With the exception of lower preschool attendance in non-Aboriginal boys than non-Aboriginal girls, there were no significant differences between boys and girls on other measures (Table 3).

In contrast, for both Aboriginal and non-Aboriginal children, those from non-English speaking backgrounds scored lower on almost all measures (Table 3). One exception was Year

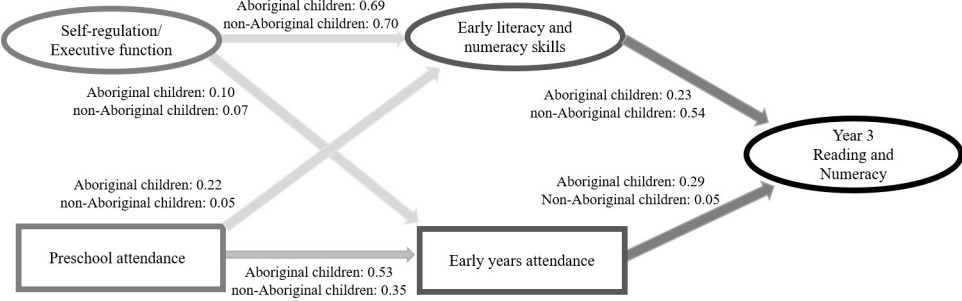

**Fig 3. The mediation model with early literacy/numeracy skills and early years attendance as mediators.** All path coefficients are standardised.

**Table 3. Standardised beta coefficients from the structural equation modelling with Year 3 academic outcome (i.e. mediation model).**

| Aboriginal children | | | | | |
|---|---|---|---|---|---|
| | Self-regulation and executive function | Preschool attendance | Early literacy/ numeracy skills | Early years attendance | Year 3 reading and numeracy |
| **Male** | -0.23*** | 0.02(NS) | 0.03(NS) | 0.00(NS) | -0.01(NS) |
| **non-English speaking** | -0.18*** | -0.37*** | -0.08** | -0.16*** | -0.21*** |
| **Remoteness** | -0.02(NS) | 0.01(NS) | -0.10*** | -0.10*** | -0.11** |
| **Socio-economic status** | 0.04(NS) | 0.10*** | -0.01(NS) | 0.04(NS) | 0.09** |
| **Self-regulation and executive function** | | | 0.69*** | 0.10*** | 0.03(NS) |
| **Preschool attendance** | | | 0.22*** | 0.53*** | -0.04(NS) |
| **Early literacy/numeracy skills** | | | | | 0.23*** |
| **Early years attendance** | | | | | 0.29*** |
| non-Aboriginal children | | | | | |
| | Self-regulation and executive function | Preschool attendance | Early literacy/ numeracy skills | Early years attendance | Year 3 reading and numeracy |
| **Male** | -0.19*** | -0.05* | -0.01(NS) | 0.00(NS) | 0.04(NS) |
| **non-English speaking** | -0.14*** | -0.11*** | -0.10*** | -0.05* | 0.03(NS) |
| **Remoteness** | 0.06* | 0.02(NS) | -0.08** | -0.05(NS) | 0.06* |
| **Socio-economic status** | 0.08** | 0.12*** | -0.02(NS) | 0.02(NS) | 0.14*** |
| **Self-regulation and executive function** | | | 0.70*** | 0.07** | 0.02(NS) |
| **Preschool attendance** | | | 0.05* | 0.35*** | -0.01(NS) |
| **Early literacy/numeracy skills** | | | | | 0.54*** |
| **Early years attendance** | | | | | 0.05* |

3 academic outcomes for non-Aboriginal children who showed no evidence of lower scores for non-English speaking background children ($\beta = 0.03$, $p > 0.05$), compared to English speaking, non-Aboriginal children in Year 3. This suggests that for non-Aboriginal children the impact of a non-English speaking background has dissipated by the time they undertake their Year 3 reading/numeracy assessment ($\beta = 0.03$, $p > 0.05$). In contrast, non-English speaking background continues to influence the assessed academic outcomes for Aboriginal children ($\beta = -0.21$, $p < 0.001$). The effect of non-English speaking background on all measures, except early literacy/numeracy skills, appears to be greater for Aboriginal children than non-Aboriginal children, particularly for preschool attendance (Aboriginal: $\beta = -0.37$, $p < 0.001$; non-Aboriginal: $\beta = -0.11$, $p < 0.001$) and Year 3 reading/numeracy (Aboriginal: $\beta = -0.21$, $p < 0.001$; non-Aboriginal $\beta = 0.03$, $p = 0.34$).

For both Aboriginal and non-Aboriginal children, remoteness was associated with lower early literacy/numeracy skills (Aboriginal: $\beta = -0.10$, $p < 0.001$; non-Aboriginal: $\beta = -0.08$, $p < 0.001$). For the Aboriginal children, there was also strong evidence of the negative association between remoteness and early years attendance ($\beta = -0.10$, $p < 0.001$), and a negative association between remoteness and Year 3 reading/numeracy score ($\beta = -0.11$, $p < 0.001$). In contrast, for the non-Aboriginal children, there was some evidence that Year 3 reading/ numeracy score was positively related to remoteness ($\beta = 0.06$, $p < 0.05$).

For both Aboriginal and non-Aboriginal children, higher socio-economic status was associated with higher preschool attendance (Aboriginal: $\beta = 0.10$, $p < 0.001$; non-Aboriginal: $\beta = 0.12$, $p < 0.001$) and higher Year 3 reading/numeracy scores (Aboriginal: $\beta = 0.09$, $p < 0.01$; non-Aboriginal: $\beta = 0.06$, $p < 0.05$).

**Pathway analysis for subgroups.**  S7 Table in S1 File shows the direct and indirect pathways in the mediation analysis for both Aboriginal and non-Aboriginal children described by sex, English speaking background, remoteness and socio-economic status, respectively.

Among the Aboriginal children, the direct effect of early literacy/numeracy skills on Year 3 reading and numeracy outcomes was weaker for children with non-English speaking backgrounds (β = 0.09) than children from English speaking backgrounds (β = 0.61) ($\chi2$ (1) = 15.03, $p$ <0.001). The indirect effect of self-regulation and executive function on Year 3 reading and numeracy was also weaker for children with non-English speaking backgrounds (β: 0.11) than children with English speaking backgrounds (β = 0.47), ($\chi2$ (1) = 12.17, $p$ <0.001).

Interestingly, there was evidence of stronger direct effects of early years attendance for Aboriginal children with non-English speaking backgrounds (β = 0.37) than Aboriginal children with English speaking backgrounds (β = 0.14), ($\chi2$ (1) = 3.72, p = 0.054).

Similar patterns were also observed for Aboriginal children living in remote (i.e. compared to urban-residing children) and most disadvantaged regions (i.e. compared to children not living in most disadvantaged regions) (S7 Table in S1 File).

## Discussion

Two key discussion points arise from the findings of our study. The first point is the positive effect of self-regulation and executive function skills on early years academic outcomes for all children in the NT, which is consistent with other international and Australian studies [4, 42–45]. This finding speaks to the need to elevate the importance of these foundational skills in policies, programs and data collection as discussed below. Secondly, our finding that Aboriginal and non-Aboriginal children experience different pathways for the effects of self-regulation and executive function on academic outcomes, demands further investigation to culturally, linguistically and contextually differentiate programs and policies to support these skills appropriately and responsively in the current Australian education context. The pending implementation of the Closing the Gap 2020 recommendations to extend access to two years of quality preschool for Aboriginal children, creates an urgency to better understand the key principles or approaches that are needed to achieve contextually responsive and effective programs [6].

### Self-regulation and executive function

**Importance of self-regulation and executive function for all children.**  Studies across a range of contexts are increasingly paying attention to the positive effect of self-regulation and executive function skills on pathways to academic outcomes, school or academic engagement and well-being. Our study's finding of a positive effect of self-regulation and executive function skills on early years academic outcomes for all children in the NT is particularly important given the increasing rates of school disengagement as evidenced by declining attendance and achievement. There is an urgent need to better connect learners and their school learning meaningfully and authentically with their worlds [46, 47]. Research establishing ecological models for the social determinants of health and learning [48] are now enhanced by the mapping of complex psychosocial factors that contribute to inequalities in health and education outcomes in populations [49]. Added to this is the emerging evidence of the ways in which early life experiences of toxic stress impact children and young people's genetic coding for stress regulation [50]. The evidence base underpinning our theory of change comes from the international and national literature mapping the early life experiences that contribute to academic outcomes of children through social and emotional capabilities. The complexity of drivers in the literature has not been fully explored in our study. Rather, we have aimed to establish the extent to which self-regulation and executive function skills feature in the

pathway to academic outcomes. An important future analysis will be to explore the available data for relationships between preschool attendance and self-regulation and executive function.

**Policy and programs.** In recent years policy agendas have typically paid more attention to the contribution of attendance and early literacy and numeracy on academic outcomes in the early years and longer term. Our study also emphasises the importance of attendance and early literacy and numeracy in the pathway to academic achievement, particularly for Aboriginal children. However, we know that despite large investments in school truancy programs and policies of income management which tie welfare payments to school attendance, for groups of students such as remote and Aboriginal students, disengagement has increased [47]. Although large investments have been made by several school systems in social and emotional learning, the implementation of programs is somewhat ad hoc [51]. School readiness research has long identified that safety, security and good mental health, including self-regulation are foundational to being 'ready to learn' in formal settings. Increasingly, research is examining the relationships between poverty and other contributors to disadvantage on social and emotional health and engagement with early learning or schooling. Hence the importance of system level policies that address universal and targeted needs in the selection and implementation of programs.

Our findings underscore the importance of including Social and Emotional Learning (SEL) in the strategic policy priorities for the NT Department of Education. The NT Department of Education's SEL package aims to develop students' self-regulatory and executive function skills including resilience, management of emotions, behaviours and relationships with others as foundational skills for learning throughout the early years and beyond [52]. In a recent review of social and emotional learning, distinct cultural differences were evident in self-regulatory practices particularly between collectivist cultures and individualistic cultures such as found in the Aboriginal and non-Aboriginal cultures of the Northern Territory [51]. The implication of our findings for different pathways to academic outcomes for Aboriginal and non-Aboriginal children, is that whilst schools are an excellent place to deliver a supportive curriculum and provide opportunity for children and young people exercise their learning, effective curriculum may need to be more responsive to the cultural differences in values and beliefs about social, emotional and relational skills. In a related study (pending publication), we found that it is essential to support teachers with professional learning about teaching self-regulation and executive function skills for their sense of self-efficacy.

**Implementation of contextualized programs.** In the Northern Territory, preschool has been delivered using a variety of service models due to the distribution of the population and the diversity of cultural and social contexts. These alternative service delivery models, including co-located and standalone preschools, multi-level early year's classes, mobile early childhood education services, distance education (School of the Air), and satellite programs (where transporting children to the nearest primary school was not feasible), have a demonstrated relationship with outcomes [1]. Further, the Productivity Commission Report in 2020 commented on the continuing fragmentation of early childhood services resulting in ongoing gaps and duplication of funding to services which often did not address community interests of needs [53]. Further to the issue of policy implementation is the importance of implementation of place-based strategies such as an integrated services model for early childhood services and Aboriginal community and health services [53, 54]. National policy reforms and bilateral funding agreements in 2008 by the Council of Australian Governments included a roll-out of such integrated services which are only just coming to fruition despite the strong evidence base from Australian and Canadian approaches [55–59].

Much research internationally and in Australia has identified culture, language and mobility as barriers to accessing early childhood, schooling and health services for Aboriginal people and other marginalised populations [60–63]. Services which are most effective or responsive to Aboriginal people in socio-economically or geographically disadvantaged communities are integrated and comprehensive [47, 64]. Further, these services are designed with community or Aboriginal organisations for empowerment and cultural capital or continuity. They are staffed by highly (and culturally) competent personnel to meet the complex and multiple issues faced by families and communities living in disadvantage often compounded by mental health, depression and substance dependencies or abuse [55, 65–70]. The requirement for place-based or community designed services is a key component of the Closing the Gap 2020 agenda to address the health and education inequities for First Nation Peoples of Australia. This incorporates the need for empowerment and cultural capital in services that are aligned with the value, beliefs and needs of the community.

**Data collection and measures.** The adage that 'we treasure what we measure' is a current consideration in raising and addressing the importance of curriculum and pedagogies that include self-regulation and executive function skills. There is increasing attention in the Australian literature and policy space on better measures of these skills [45, 71]. In the NT, although there is a substantial investment in social and emotional wellbeing curriculum and professional learning, there is no system-level collection of data to establish the efficacy of these programs, or to contribute to continuous improvement in educational and developmental outcomes in primary and secondary schools. Studies investigating self-regulation and executive function in the early years have depended on a range of indicators in the AEDC.

In 2009, Australia became the first country in the world to collect national data on the developmental outcomes of all children starting school through the AEDC [72]. While this national data collection of early childhood indicators is to be celebrated, there would be significant value in extending this to middle childhood and early adolescence. Currently, New South Wales and South Australia implement population-level assessments of children's social-emotional development and wellbeing in the middle years. The New South Wales (NSW) Child Development Study assesses children's mental health and well-being at approximately 11 years of age [73]. The South Australian Department of Education developed the 'Wellbeing and Engagement Collection' for school students aged 8–12 years, in collaboration with the developers of the Canadian Middle Years Development Indicator since 2018 [74]. This survey provides schools, the community and government an insight into the non-academic needs of students for engagement and success. Such data collections could serve to give a better longitudinal picture of children and young people's social and emotional learning and strengths that facilitate children's 'readiness to learn', particularly the ever-increasing exposure to stresses and trauma [6]. Further, the cultural and linguistic contexts of remote Aboriginal children may require more nuanced approaches to data collection and measures of self-regulation and executive function [75].

**Different pathways for Aboriginal and non-Aboriginal children.** This study's findings that Aboriginal and non-Aboriginal children in the NT follow different pathways towards academic achievement is understood in relation to the distinct strengths and assets available to children living in urban and non-urban settings which also follows these two cohorts. Firstly, a comprehensive body of research now exists on the role of preschool participation for all children and especially children living in disadvantage. There is also a substantial body of literature on this disadvantage being in relation to the cultural, linguistic and world view privileged in schools that is usually not inclusive of Aboriginal children's experiences. In Australia, preschools as part of early childhood system are predicated on social justice and bridging

structural equality. Hence, the importance attached to preschool attendance in policy and reform agenda can be understood.

Our previous study found that for Aboriginal children, the pathway to Year 3 NAPLAN outcomes is also mediated by their school attendance (from Years 1 to 3), whilst non-Aboriginal children's pathway is mediated through school readiness skills [1]. However, the direct effect of early years attendance was stronger amongst Aboriginal children. For Aboriginal children, early literacy/numeracy and self-regulation and executive function had a weaker effect on Year 3 reading/numeracy by comparison with non-Aboriginal children, and was influenced by non-English speaking backgrounds, living in remote and socio-economically disadvantaged communities. As expanded on below, these four factors (i.e. attendance, non-English speaking background, remoteness and socio-economic disadvantage) are complex and inter-related. Some clear implications and explanations can be made about attendance and learning English as an additional language. However, the complexity and inter-relatedness of these four factors, and particularly remoteness and socio-economic disadvantage as community level factors, is only just beginning to be understood from Aboriginal theoretical and methodological perspectives. This emerging literature offers hope in gaining better appreciation of the strengths and assets available to Aboriginal children living in remote communities across Australia and other parts of the world. Additionally, there is an emerging field of research on the intergenerational impact of colonisation on misdirected policy and programs, and how to co-design responses that are affirming of culture, language and identity particularly in addressing issues of community safety and the high levels of stress [6, 75–77].

**Preschool participation and school attendance.** Our findings of the positive impact of preschool attendance on Year 3 reading/numeracy for Aboriginal children, highlights the significant benefits of regular preschool attendance. This aligns with the inclusion of enrolment and participation in early childhood education as one of the new Closing the Gap targets relating to Aboriginal educational outcomes (increasing to 95% by 2025) [23]. Previous NT study provides *"encouraging empirical evidence for increased preschool attendance of Aboriginal children being associated with increased early year school attendance rates and thus better NAPLAN achievement outcomes"* [1]. The same study also found the greater effect of preschool attendance on early school attendance rates for Aboriginal students than non-Aboriginal students in the NT. The stronger relationship between preschool attendance and Year 3 reading and numeracy for Aboriginal children by comparison with non-Aboriginal children is a function of learning English as a foreign language in many remote communities. The aspirational goal of early childhood system is to achieve equity across all life outcomes and this is reflected in the 2020 Closing the Gap Partnership Agreement. Particular attention is given to how equity in outcomes requires differentiated early childhood programs for Aboriginal and Torres Strait Islander communities. This includes more holistic services, bilingual and culturally inclusive educators. In many community contexts where families may be managing multiple and complex issues, *"children's preschool participation helps parents to build the habit of structuring a typical day around their children's school routine"* [1]. It is possible to design early childhood education provision that recognises the universal benefit for all children, while also taking into account that some children benefit more or require additional support to achieve the same outcome. Known as the proportionate universalism approach [78–80], every child would receive a baseline level of preschool provision, and vulnerable children and families would receive extra support. For example, in the 2008 Coalition of Australian Governments' reform agenda, it was proposed that Aboriginal children would have access to two years of preschool to address a number of areas of need. This did not come to fruition. In the NT, children do have access to publicly provided preschool for a minimal and voluntary parent contribution —"the majority of preschool programs (94%) were delivered free of charge for children aged

from 4 years in provincial and remote areas and from 3 years in very remote areas by the NT government" [1].

Our study also demonstrates the importance of school attendance from Transition to Year 2 on academic outcomes of Aboriginal children. While the importance of preschool is widely acknowledged, a previous NT study indicated that the initial benefits of preschool can easily 'fade out' unless they are reinforced by regular attendance and effective engagement with school learning in the early years of primary school. Silburn (2016) highlighted the necessity of policy and services supporting children's transition into formal school learning extending through to at least Year 3. However, the factors contributing to non-attendance have been demonstrated to be complex and must be considered with respect to the conditions and policies determined by a range of agencies related to housing, welfare, health and justice [27, 49, 81].

As demonstrated in previous studies, the multiplicity of socio-demographic and early life health factors influencing Aboriginal children's school attendance, highlights the extent to which whole-of-government policy investments are needed to better address housing overcrowding, maternal and child health, early childhood education and care, parental education and employment. For example, it was found that the most important factor of school attendance in Year 1 amongst NT Aboriginal children is living in a community with overcrowded housing (i.e. associated with 35 days (seven school weeks) absence from school) [1]. A separate study reported the adverse impact of hearing impairment (resulting from middle ear diseases) on school attendance of Aboriginal children in Year 1 [82]. Hence, policies and service initiatives to 'close the gap', will only be effective if meaningful progress can be made in addressing Aboriginal children's disproportionate exposure to disadvantage [1], especially amongst those children living in remote and disadvantaged regions.

**Non-English speaking background.** Our study highlights the different academic challenges for children with non-English speaking backgrounds amongst Aboriginal and non-Aboriginal population. For non-Aboriginal children, the impact of a non-English speaking background has dissipated by the time they undertake Year 3 reading/numeracy tests. In comparison, for Aboriginal children, this impact continues at Year 3 and likely, beyond. Previous studies suggested that families from non-English-speaking backgrounds have additional challenges in engaging with their children's preparation for school. For some Aboriginal and non-English speaking families, despite the great value placed on their children's learning, the experience of engaging with school or teachers can be intimidating or unfamiliar, depending on past experiences, such as having limited formal education themselves, and language barriers. Moreover, teachers may sometimes be unaware or unresponsive to parents' perceptions, the power relationships, and cultural barriers in such diverse contexts [83–85].

**Importance of identify, culture and country connections.** Consistent with the premise of the present study and the findings of the Accelerated Literacy Program [86], we surmise that it is likely children's success in school learning is underpinned by a set of more foundational 'ready to learn' skills that we identify as self-regulation and executive function, when entering into schools. Our finding that Aboriginal and non-Aboriginal children are likely to experience different pathways, places great importance on further exploration of culturally, linguistically and contextually appropriate and responsive SEL programs [6, 75]. Our findings suggest that policies or programs to improve early development and educational outcomes of Aboriginal children must recognise that a single intervention might not be sufficient, and that 'contextually appropriate multi-model interventions in partnership with local communities and their stakeholders' are required [87]. Recent research with NT urban schools found that children need safe and supportive conditions at school for self-regulation and being ready to learn [88]. Further studies are required to examine the program responses that could increase

children's sense of safety and supportive classrooms in regional, remote and very remote contexts, which we suggest are likely to increase attendance and readiness to learn academics [75, 77].

Systemic policy issues that overlook the relationship between language, culture and identity (including learning on country) can also impact learners, families and communities [6]. Many NT remote communities are contexts in which children are learning English as a foreign language, which are considerably different to context in which English is an additional language [1, 6, 89]. Despite two major reviews of NT Aboriginal education (in 1999 and 2013 respectively), systemic issues still prevail. The 1999 review recommendations supported the *"goal of Indigenous parents and community members for their children to develop English language, literacy and numeracy skills while maintaining their own language, cultural heritage and Indigenous identity"* [55]. In the 2013 review, some recommendations were viewed by observers as 'contentious' [90]. One contentious recommendation included adopting direct instruction curricula for English language skills and knowledge for 'success in the western education system' [91] at the exclusion of other bilingual approaches [89, 92]. Overlooked in the application of this recommendation, is the convergence of evidence about the need for local and contextualised pedagogic approaches that affirm identity and cultural connection by building on first language and ways of knowing [90, 93]. Closer scrutiny of the programs experienced by children is required to understand the relationships with the prevailing NAPLAN outcomes and declining attendance particularly whether conditions for cultural safety and support for self-regulation are present [22].

## Strengths and limitations of study

The main strength of the study is its use of population-level linked data which has comprehensive coverage and representation of the study population. The analysis was also stratified by Aboriginal status resulting in findings that can inform culturally relevant responses and may offer insights to analogous populations.

Limitations include the study cohort being only children who attended public schools, and the analysis being unable to make strong causal claims about the directional nature of the relationships between the different indicators. This study also used a narrow criteria for school success by using reading and numeracy scores at Year 3 NAPLAN [94]; this study is unable to investigate other to other positive educational outcomes [94] (e.g. well-being, aspirations, participation, identities, relational) due to data limitations. Finally, the data available to and used in this study did not include other important factors that may influence or modify the outcomes, such as parental involvement in their children's learning prior to or during preschool, the quality of preschool programs attended or the learning environment (e.g. peer-effects, classroom size, student-teacher ratio or teachers' teaching style).

## Conclusions

With the implementation of the Closing the Gap 2020 recommendations, there is an urgent need to better understand self-regulation and executive functions as contributing factors to positive educational outcomes for children living in both urban and remote settings. This study had access to linked data of preschool attendance, AEDC, early years attendance and NAPLAN scores, and so was able to provide a basic understanding of the pathways to early academic achievements for both Aboriginal and non-Aboriginal children in the NT. This study acknowledges that NAPLAN is a narrow criterion for school success. Due to data limitations, this study does not provide insights into the pathways to other important positive schooling outcomes (e.g. well-being, aspirations, participation, identities, relational). Currently

in Australia, only AEDC, NAPLAN, school enrolment and attendance data are collected nationally in the early childhood and primary education setting. The current study forms the basis for further investigation into self-regulation and executive function as contributing factors to positive educational outcomes for both Aboriginal and non-Aboriginal children in the NT and across Australia. It suggests the need for more attention to self-regulation and executive function in national data-collection.

Despite the limitations, our study offers valuable insights to better understand the contribution of early foundational skills that comprise self-regulation and executive function to positive educational outcomes in different populations. The results demand further investigation to culturally, linguistically and contextually differentiated programs and policies in the current Australian education context. Our study confirms the expected importance of self-regulation and executive functioning skills for all children but suggests there are different pathways for Aboriginal and non-Aboriginal children in the NT. Our study suggested the importance of preschool and early years attendance in the pathway to academic achievement, particularly for Aboriginal children. Further, these results reflect the distinct population profile of the NT with a majority of Aboriginal children with language backgrounds other than English, living in geographically remote communities (i.e. 75%) and with substantial disadvantaged subgroups of children from rural and remote backgrounds in the major centres who have poor access to services, different from other Australian and international jurisdictions. There are potentially cultural or linguistic assets and strengths that contribute to self-regulation and executive function as foundational skills for academic learning that are not recognised in the current tools.

The complex inter-relatedness of school attendance, remoteness, non-English speaking background and socio-economic status on the pathway for self-regulation and executive function skills demand attention in the design of effective policies and programs. Policy makers and educators must recognise that the factors contributing to non-attendance are complex, hence the solutions require multi-sectoral collaboration in place-based design for effective implementation, particularly for early childhood experiences. Given the importance of self-regulation and executive function for foundational skills, and readiness for academic engagement, there is a pressing need to better understand how current policies and programs enhance children and their families' sense of safety and support to nurture these skills.

## Supporting information

**S1 File. S1—S7 Tables.**
(DOCX)

## Acknowledgments

The authors would like to thank Professor Sven Silburn for his invaluable input in reviewing a draft of this manuscript. The authors would also like to acknowledge the support by the Northern Territory Departments of Health; Education; Territory Families, Housing and Communities; Attorney General and Justice; Chief Minister and Cabinet; Treasury and Finance; and Police, Fire and Emergency Services, through the Child and Youth Development Research Partnership (CYDRP). We also thank the many data custodians who have assisted with the retrieval, preparation and release of the research datasets, and the staff of the SA NT DataLink data integration authority for their technical and administrative assistance in the linkage of datasets. The views expressed in this publication are those of the authors and not necessarily those of the NT government departments who have supported the study.

This paper uses data from the Australian Early Development Census (AEDC). The AEDC is funded by the Australian Government Department of Education, Skills and Employment. The findings and views reported are those of the author and should not be attributed to the Department or the Australian Government.

## Author Contributions

**Conceptualization:** Vincent Yaofeng He, Georgie Nutton.

**Data curation:** Vincent Yaofeng He.

**Formal analysis:** Vincent Yaofeng He.

**Funding acquisition:** Georgie Nutton.

**Investigation:** Vincent Yaofeng He.

**Methodology:** Vincent Yaofeng He, Georgie Nutton.

**Project administration:** Georgie Nutton.

**Visualization:** Vincent Yaofeng He.

**Writing – original draft:** Vincent Yaofeng He, Georgie Nutton, Amy Graham, Jiunn-Yih Su.

**Writing – review & editing:** Vincent Yaofeng He, Georgie Nutton, Amy Graham, Lisa Hirschausen, Jiunn-Yih Su.

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
