## [Decision Letter · Decision Letter 0]

1 Jul 2021

PONE-D-21-15913

Pathways to school success: Self-regulation and executive function, preschool attendance and early academic achievement of Aboriginal and non-Aboriginal children in Australia’s Northern Territory

PLOS ONE

Dear Dr. He,

Thank you for submitting your manuscript to PLOS ONE. I have now heard from three extremely knowledgeable reviewers who work in this area. All three have recommended major revision. I have read the draft and given my interest in this topic and its policy relevance. I agree with their assessment and I want you to address their concerns and I look forward to seeing a revised draft.

While revising, I want you to focus on:

1. Measurement error concerns.

2. Concerns around data.

3. Ways to improve empirical strategy - around omitted variable bias (see R2).

4. All referees have some concerns around the indices. I think you should try to add robustness on other ways to create these indices, disaggregate the indices -- this can go in Appendix as robustness.

5. One of the referee also has some thoughts on mediation. I will encourage you to think about it. In addition, you should be more clear about the underlying mechanisms. 

6. Be more clear about the contributions of this study.

Thank you for submitting your manuscript to PLOS ONE. After careful consideration, we feel that it has merit but does not fully meet PLOS ONE’s publication criteria as it currently stands. Therefore, we invite you to submit a revised version of the manuscript that addresses the points raised during the review process. Overall this is an excellent paper and I very much look forward to seeing the revised draft.

We look forward to receiving your revised manuscript.

Kind regards,

Nishith Prakash, Ph.D.

Academic Editor

PLOS ONE

Journal Requirements:

Reviewers' comments:

Reviewer's Responses to Questions

**Comments to the Author**

1. Is the manuscript technically sound, and do the data support the conclusions?

Reviewer #1: Partly

Reviewer #2: Yes

Reviewer #3: Partly

2. Has the statistical analysis been performed appropriately and rigorously? 

Reviewer #1: Yes

Reviewer #2: Yes

Reviewer #3: I Don't Know

3. Have the authors made all data underlying the findings in their manuscript fully available?

Reviewer #1: Yes

Reviewer #2: Yes

Reviewer #3: Yes

4. Is the manuscript presented in an intelligible fashion and written in standard English?

Reviewer #1: Yes

Reviewer #2: Yes

Reviewer #3: Yes

5. Review Comments to the Author

Reviewer #1: Referee Report on “Pathways to School Success: Self-Regulation and Executive Function, pre-school attendance and early academic achievement of Aboriginal and non-Aboriginal children in Australia’s Northern Territory”

I. Summary and Contribution

This paper studies the pathways through which behavioral characteristics such as self-regulation, executive function as well as attendance affect early childhood academic achievement. It is well-known in the education literature that early childhood outcomes set the tone for an individual’s later life outcomes and any faltering in early childhood can be hard to compensate for. In this regard, this paper makes an important contribution. This paper finds that between Aboriginal children and non-Aboriginal children, the effect of self-regulation and executive function on academic scores is mediated through early literacy/numeracy scores in non-Aboriginal children but through both school attendance and early literacy/numeracy scores for Aboriginal children. The results contribute to a better understanding of how historic exploitation of a population – the Aboriginals can manifest in childhood education outcomes. Another striking result in this paper is that the impact of non-English background lingers for Aboriginal children but dissipates for non-Aboriginal children. This result adds to the body of evidence on how children from disadvantaged backgrounds find it harder to overcome cultural differences.

II. Comments

1. The score for self-regulation and executive function is calculated by adding the answers to around 90 questions. But an index/score created by addition implies an implicit assumption that these characteristics are substitutable. A low score in one domain by an individual can be compensated by a high score in another domain. In such a score design, how well does it capture an individual’s self-regulation and executive function? A score design which is substitutable within domain but not across domains can be a better measure of individual capabilities. Alternatively, one can employ exploratory factor analysis/dimensionality reduction techniques to construct a better score.

2. The authors mention that aboriginal status was predicted using an algorithm. First, the details of the algorithm employed, and a discussion of its prediction accuracy is needed. Since the most striking results of this paper is the difference of outcomes and pathways between Aboriginals and non-Aboriginals, it is important to understand how well the algorithms predicts Aboriginal status to begin with. Second, every algorithm no matter how well trained, has a prediction error. When a prediction is incorporated into a model especially regression models, the prediction error passes on as measurement error. Here, it is measurement error in the covariates which will affect the estimated coefficients (unlike measurement error in the dependent variable which can be absorbed in the error term). Measurement error problems rarely have good solutions. A discussion of how much of a concern it can be, stemming from what the prediction error in predicting Aboriginal status was, will strengthen the paper.

3. The result of this paper (importance of self-regulation and executive function in childhood academic achievement) are expected and intuitive result. Confirmation of expected results are indeed an important contribution to the literature. However, this paper warrants a more rigorous discussion of the specific contribution to our understanding of drivers of early childhood academic achievement. How does the effect sizes of these two drivers compare to that of other factors studied in the literature? The authors mention that previous literature have not looked at these drivers. But why specially these drivers are important to look at is not discussed well. A discussion connecting these results to broader behavioral literature will greatly improve this paper.

4. What implications do the results have for future policy-making? The conclusion only suggests that policy-makers be cognizant of the differences between Aboriginal children and non-aboriginal children. Examples of specific policies that can be created (from other countries perhaps) will make the paper stronger.

Reviewer #2: Regarding Q1, I found the manuscript technically sound, using appropriate data.

Regarding Q2, statistical analysis is both appropriate and rigorous. I have suggested a few more in my attached comments.

Regarding Q3, I selected "yes" because I believe authors would make replication files available following the journal data policy.

Regarding Q4, I found the manuscript to be in an intelligible fashion and written in standard English.

Please see the attachment for detailed comments.

Reviewer #3: Thanks for inviting me to review. My main reflections are as follows:

1. The topic being studied is of great importance. Understanding better the precise ways in which investments in early childhood can improve student skills is vital to the design of better public policy investments in this area. Within this broad umbrella, exploring the precise specific constraints faced by children from marginalized communities is extremely important.

2. I also appreciate the care the authors have taken to use well-established administrative data and bring different pieces of the data together to create a large representative sample

3. One point of confusion is the relationship between three different factors: (i) preschool attendance; (ii) self-regulation and executive function; and (iii) early academic achievement (literacy/numeracy). As I understand it, the results seem to suggest that pre-school attendance mediates the relationship between (ii) and (iii) for Aboriginal children but not so much for non-Aboriginal children. I would have really liked to see a clear exploration of the impact of pre-school attendance on self-regulation and executive function – and how this varies between Aboriginal and non-Aboriginal children. I am not sure this aspect comes out clearly in the paper. Once we are clearer on this relationship, I believe we can interpret the broader results on how the pre-school attendance is mediating the relationship between self-regulation and executive function and early academic achievement – and how it varies between Aboriginal and non-Aboriginal children. More fundamentally, I think the authors should map out a clear theory of change between the main relationships modelled.

4. Another thing that bothered me was that we don’t see the extent to which the likelihood of being remote and non-English speaking varies between Aboriginal and Non-Aboriginal children.

5. Having said that, some of the key policy recommendations seem sound. I just think the authors could do more to persuade the reader that the way the relationships are being examined makes sense.

6. PLOS authors have the option to publish the peer review history of their article (what does this mean?). If published, this will include your full peer review and any attached files.

Reviewer #1: No

Reviewer #2: No

Reviewer #3: No

---

## [Author Response · Author response to Decision Letter 0]

12 Oct 2021

We thank all the reviewers for their acknowledgement of the need to better understand the impact of and the relationships between the various factors in our unique context and diverse population including the historic legacy of colonisation; socio-economic and political disadvantage. Their insights and suggestions have assisted the authors greatly. We have considered each comment and provide the following responses, along with the proposed revisions to the manuscript. We have included the proposed changes in the revised manuscript .

Reviewer 1

R1C1: 

The score for self-regulation and executive function is calculated by adding the answers to around 90 questions. But an index/score created by addition implies an implicit assumption that these characteristics are substitutable. A low score in one domain by an individual can be compensated by a high score in another domain. In such a score design, how well does it capture an individual’s self-regulation and executive function? A score design which is substitutable within domain but not across domains can be a better measure of individual capabilities. Alternatively, one can employ exploratory factor analysis/dimensionality reduction techniques to construct a better score.

In our main analysis, we used structural equation model in which we used scores from nine sub-domains as 9 indicators (aka observed variables) to form a single latent construct for self-regulation and executive function. Therefore, for the main analysis we did not use an aggregated score for self-regulation and executive function by adding up the scores from all nine sub-domains. In the data-analysis subsection of the method section, we have stated clearly that we first conduct a confirmatory factor analysis, and then used structural equation model.

We apologise for the confusion caused. For our descriptive analysis (Table 1), we have originally used the aggregated scores for ‘self-regulation and executive function’ latent construct (sum of nine sub-domains) and ‘early literacy/numeracy skills’ latent construct (sum of three sub-domains). The purpose of Table 1 is used to illustrate the differences in the scores for ‘self-regulation and executive function’ and ‘early literacy/numeracy skills’ amongst the different sub-groups. However, the aggregated scores were not used in the main analysis that involved confirmatory factor analysis and structural equation model.

To improve the clarity and avoid confusion to the readers, we have made the two major changes. The first change occurred in Figure 2 and Figure 3, in which we used the conventional graphical representations of the structural equation model, by using oval shape for the ‘self-regulation and executive function’ and ‘early literacy/numeracy skills’ latent construct (to signify latent/unobserved variables), and square shape for the other observed variables. The second change occurred in Table 1, in which rather than using the two aggregated scores for ‘self-regulation and executive function’ and ‘early literacy/numeracy skills’, we reported the proportion of children developmental vulnerable on each of the AEDC sub-domains in the self-regulation and executive function latent construct and early literacy/numeracy latent construct. We have revised the main text in the ‘measures’ sub-section in the ‘method’ section below:

In the descriptive analysis (i.e. Table 1 and S2 Table), the proportion of children identified as developmentally vulnerable in each of the sub-domains (i.e. scored in the bottom 10% of the national AEDC population) was presented (1). In the SEM, the standardised score from each of the nine AEDC sub-domain was used as manifest indicator variables for the latent construct ‘self-regulation and executive function’. 

R1C2: 

The authors mention that aboriginal status was predicted using an algorithm. First, the details of the algorithm employed, and a discussion of its prediction accuracy is needed. Since the most striking results of this paper is the difference of outcomes and pathways between Aboriginals and non-Aboriginals, it is important to understand how well the algorithms predicts Aboriginal status to begin with. Second, every algorithm no matter how well trained, has a prediction error. When a prediction is incorporated into a model especially regression models, the prediction error passes on as measurement error. Here, it is measurement error in the covariates which will affect the estimated coefficients (unlike measurement error in the dependent variable which can be absorbed in the error term). Measurement error problems rarely have good solutions. A discussion of how much of a concern it can be, stemming from what the prediction error in predicting Aboriginal status was, will strengthen the paper.

We apologise for the confusion caused. In our study, we are not using a predictive model, therefore the discussion of prediction error is not relevant in the discussion. However, to improve the clarity, we have revised the main text relating to the derivation of the variable for Aboriginal status.

“In our study, the Aboriginal status variable determined with an algorithmic approach using the same Aboriginal status variable in every dataset of the CYDRP data repository based on their respective demonstrated levels of accuracy, firstly using health datasets, followed by child protection data, and then education and youth justice records (2). This hierarchy of accuracy was based on systematic evaluation of the completeness and quality of each dataset referenced to health records (i.e. hospital data) for which an audit, in 2011, demonstrated 98% consistency between recorded Aboriginal status and patient interview (3). The aforementioned approach is described in detail elsewhere (2) and is consistent with best practice guidelines involving data linked from two or more datasets (4),

R1C3: 

The result of this paper (importance of self-regulation and executive function in childhood academic achievement) are expected and intuitive result. Confirmation of expected results are indeed an important contribution to the literature. However, this paper warrants a more rigorous discussion of the specific contribution to our understanding of drivers of early childhood academic achievement. How does the effect sizes of these two drivers compare to that of other factors studied in the literature? The authors mention that previous literature have not looked at these drivers. But why specially these drivers are important to look at is not discussed well. A discussion connecting these results to broader behavioral literature will greatly improve this paper

The discussion now elaborates on the broader literature’s theory and empirical evidence for the assumptions made in this study. This study is only concerned with the relationships between SR-EF and academic outcomes and the inclusion of attendance in pre-school and early years is a known and well established factor for children in the NT. Other important relationships with the patterns of attendance and the development of SR-EF at age 5 are the subject of our next investigations. 

We have added the following paragraph in our discussion section:

“Our study’s finding of a positive effect of self-regulation and executive function skills on early years academic outcomes for all children in the NT is particularly important given the increasing rates of school disengagement as evidenced by declining attendance and achievement. There is an urgent need to better connect learners and their school learning meaningfully and authentically with their worlds (5, 6). Research establishing ecological models for the social determinants of health and learning (7) are now enhanced by the mapping of complex psychosocial factors that contribute to inequalities in health and education outcomes in populations (8). Added to this is the emerging evidence of the ways in which early life experiences of toxic stress impact children and young people’s genetic coding for stress regulation (9). The evidence base underpinning our theory of change comes from the international and national literature mapping the early life experiences that contribute to academic outcomes of children through social and emotional capabilities. The complexity of drivers in the literature has not been fully explored in our study. Rather, we have aimed to establish the extent to which self-regulation and executive function skills feature in the pathway to academic outcomes. An important future analysis will be to explore the available data for relationships between preschool attendance and self-regulation and executive function.” 

R1C4: 

What implications do the results have for future policy-making? The conclusion only suggests that policy-makers be cognizant of the differences between Aboriginal children and non-aboriginal children. Examples of specific policies that can be created (from other countries perhaps) will make the paper stronger.

We have restructured the discussion in which we present the major policy and program implications, by adding additional paragraphs. 

1.2 Policy and Programs 

In recent years policy agendas have typically paid more attention to the contribution of attendance and early literacy and numeracy on academic outcomes in the early years and longer term. Our study also emphasises the importance of attendance and early literacy and numeracy in the pathway to academic achievement, particularly for Aboriginal children. However, we know that despite large investments in school truancy programs and policies of income management which tie welfare payments to school attendance, for groups of students such as remote and Aboriginal students, disengagement has increased (6). Although large investments have been made by several school systems in social and emotional learning, the implementation of programs is somewhat ad hoc (10). School readiness research has long identified that safety, security and good mental health, including self-regulation are foundational to being “ready to learn” in formal settings. Increasingly, research is examining the relationships between poverty and other contributors to disadvantage on social and emotional health and engagement with early learning or schooling. Hence the importance of system level policies that address universal and targeted needs in the selection and implementation of programs. 

Our findings underscore the importance of including Social and Emotional Learning (SEL) in the strategic policy priorities for the NT Department of Education. The NT Department of Education’s SEL package aims to develop students’ self-regulatory and executive function skills including resilience, management of emotions, behaviours and relationships with others as foundational skills for learning throughout the early years and beyond (11). In a recent review of social and emotional learning, distinct cultural differences were evident in self-regulatory practices particularly between collectivist cultures and individualistic cultures such as found in the Aboriginal and non-Aboriginal cultures of the Northern Territory (10). The implication of our findings for different pathways to academic outcomes for Aboriginal and non-aboriginal children, is that whilst schools are an excellent place to deliver a supportive curriculum and provide opportunity for children and young people exercise their learning, effective curriculum may need to be more responsive to the cultural differences in values and beliefs about social, emotional and relational skills. In a related study (pending publication), we found that it is essential to support teachers’ with professional learning about teaching self-regulation and executive function skills for their sense of self-efficacy. 

1.3 Implementation of contextualized programs

In the Northern Territory, preschool has been delivered using a variety of service models due to the distribution of the population and the diversity of cultural and social contexts. These alternative service delivery models, including co-located and standalone preschools, multi-level early year’s classes, mobile early childhood education services, distance education (School of the Air), and satellite programs (where transporting children to the nearest primary school was not feasible), have a demonstrated relationship with outcomes (2). Further, the Productivity Commission Report in 2020 commented on the continuing fragmentation of early childhood services resulting in ongoing gaps and duplication of funding to services which often did not address community interests of needs (12). Further to the issue of policy implementation is the importance of implementation of place-based strategies such as an integrated services model for early childhood services and Aboriginal community and health services (12, 13). National policy reforms and bilateral funding agreements in 2008 by the Council of Australian Governments included a roll-out of such integrated services which are only just coming to fruition despite the strong evidence base from Australian and Canadian approaches (14-18). 

Much research internationally and in Australia has identified culture, language and mobility as barriers to accessing early childhood, schooling and health services for Aboriginal people and other marginalised populations (19-22). Services which are most effective or responsive to Aboriginal People in socio-economically or geographically disadvantaged communities are integrated and comprehensive (6, 23). Further, these services are designed with community or Aboriginal organisations for empowerment and cultural capital or continuity. They are staffed by highly (and culturally) competent personnel to meet the complex and multiple issues faced by families and communities living in disadvantage often compounded by mental health, depression and substance dependencies or abuse (14, 24-29). The requirement for place-based or community designed services is a key component of the Closing the Gap 2020 agenda to address the health and education inequities for Aboriginal and Torres Strait Island Peoples of Australia. This incorporates the need for empowerment and cultural capital in services that are aligned with the value, beliefs and needs of the community.

2.1 Preschool participation and school attendance 

Previous NT study provides “encouraging empirical evidence for increased preschool attendance of Aboriginal children being associated with increased early year school attendance rates and thus better NAPLAN achievement outcomes” (2). The same study also found the greater effect of preschool attendance on early school attendance rates for Aboriginal students than non-Aboriginal students in the NT. The stronger relationship between preschool attendance and Year 3 reading and numeracy for Aboriginal children by comparison with non-Aboriginal children is a function of learning English as a foreign language in many remote communities. The aspirational goal of early childhood system is to achieve equity across all life outcomes and this is reflected in the 2020 Closing the Gap Partnership Agreement. Particular attention is given to how equity in outcomes requires differentiated early childhood programs for Aboriginal and Torres Strait Islander communities. This includes more holistic services, bilingual and culturally inclusive educators. In many community contexts where families may be managing multiple and complex issues, “children’s preschool participation helps parents to build the habit of structuring a typical day around their children’s school routine.”(2) It is possible to design early childhood education provision that recognises the universal benefit for all children, while also taking into account that some children benefit more or require additional support to achieve the same outcome. Known as the proportionate universalism approach, every child would receive a baseline level of preschool provision, and vulnerable children and families would receive extra support. For example, in the 2008 Coalition of Australian Governments’ reform agenda, it was proposed that Aboriginal children would have access to two years of preschool to address a number of areas of need. This did not come to fruition. In the NT, children do have access to publicly provided preschool for a minimal and voluntary parent contribution—"the majority of preschool programs (94%) were delivered free of charge for children aged from 4 years in provincial and remote areas and from 3 years in very remote areas by the NT government.” 

Reviewer 2

R2C1:

You say "missing data" in line 260. What's missing exactly is not clear. If some observations are missing, can you impute their values with the Aborigine status-gender mean and check your results? For example, generate a variable with mean by Aborigine status and by gender. So, there will be four types of means. Impute missing values for Aboriginal female and male, and non-Aboriginal female and male with the respective means. If you think, this is an inferior method, please suggest why. I think readers should also know about other ways of addressing missing data.

We have added the additional explanation in our method section.

“In our analysis, there was no missing data for Aboriginal status, sex, language background, remoteness, preschool attendance, early years attendance, Year 3 NAPLAN reading scores and numeracy scores. There was only 1 record missing data for the ‘socio-economic status’ variable. Missing data for the scores in the three sub-domains (that were used to construct early literacy/numeracy skills latent construct) ranged from 3.3%-3.4%. Missing data for the scores in the nine sub-domains (that were used to construct the self-regulation and executive functioning latent construct) ranged from 3.1% to 6.3%.”

Imputing missing values with Aboriginal status-gender mean is likely to lead to biased results. In the conventional way to handle missing data, there were two major approaches with good statistical properties that produce unbiased results: maximum likelihood (ML) and multiple imputation (MI) (30). Allison (2012) stated the four reasons for the preference of ML over MI (30):

1. With MI, there is always a potential conflict between the imputation model and the analysis model. There is no potential conflict in ML because everything is done under one model.

2. The implementation of MI requires many different decisions, each of which involves uncertainty. ML involves far fewer decisions.

3. For a given set of data, ML always produces the same result. On the other hand, MI gives a different result every time you use it.

4. For a given set of data, ML always produces the same result. On the other hand, MI gives a different result every time you use it.

Our main analysis used structural equation model, and thus the robust maximum likelihood with missing values (MLMV) estimator would be the best approach to handle missing data.

As our targeted audience is the public and policy-makers, we decide not to include the rationale in the main text to avoid more confusion and distraction away from the main message that we aimed to deliver. We will include these reasons in our appendix (i.e. S3 Table).

R2C2:

On non-normality of data: most of your aggregated indices range from 0-30 or 0-90, etc. Are there zeros or are these always positive? If always positive, can you take natural logs of these variables and check your results? This is much simpler. 

In our main analysis, we did not use aggregated scores, but rather use scores from the different sub-domains as indicators (aka observed variables) to form the latent constructs for ‘self-regulation and executive function’ and ‘early literacy/numeracy skills’ respectively. In our previous manuscript, we use the aggregated scores only in the descriptive analysis in Table 2. In response to Reviewer 1, we have revised Table 1 in which we no longer report the aggregated scores, but rather report the proportion of children developmental vulnerable on each of the AEDC sub-domains in the self-regulation and executive function latent construct and early literacy/numeracy latent construct. Please see our response to Reviewer 1 Comment 1 for more details. Although aggregated indices are always positive, we did not use them in the main analysis (i.e. structural equation model).

R2C3:

Since you aggregate all components together to create variables, it is quite helpful for readers when effects are also shown using disaggregated variables. For example, for self-regulation and executive functioning, you aggregated 9 sub-domains (each ranged from 0-10) into one. 

Can you also show the effect of each domain of self-regulation and executive functioning on Year 3 learning outcomes (these can go in the appendix)? I strongly believe this will be a useful exercise because you/readers will be able to identify whether some components of self-regulation and executive functioning have direct impact on Year 3 learning and, if so, which ones. 

As mentioned previously (response to previous question and to Reviewer 1 comment 1), we used structural equation model for our main analysis. Our assumption is that ‘self-regulation and executive functioning’ latent variable is a “reflective construct” rather than a “formative construct”. Mentioned in Henseler (2021) p51, in reflective measurement model, 

“[t]he observed variables are assumed to reflect variation in a latent variable and, thereby, changes in the construct are expected to be manifested in changes in all indicators comprising the multi-item scale. Thus, the direction of causality is from the construct to the indicators”(31). On the other hand, in formative measurement model,“[f]ormative constructs occur when the items describe and define the construct” (32) (Petter, Straub, & Rai, 2007, p. 623) and“[t]he indicators are considered as immediate causes of the focal latent variable” (33) (Fassott & Henseler, 2015).

The attempt to investigate the effect of each sub-domain on Year 3 learning outcomes seem to deem ‘self-regulation and executive functioning’ as a formative construct, in which we think that is inappropriate in our context. This is because we believe that the direction of causality is from the ‘self-regulation and executive functioning’ construct to the indicators, and not in the opposite direction.

In response to this comment, in the descriptive analysis (i.e. Table 1, S3, S4, S5), we have presented the information about each sub-domain (i.e. proportion of children developmental vulnerable,%).

R2C4:

What mediation analysis model did you use? Do you check if error terms from the direct and intermediate models are correlated or independent? If correlated, how much is it biasing the effect sizes? How do you address this issue?

In our analysis involving structural equation model, we used two models: direct path model and mediation model (described in data-analysis sub-section), and did not use intermediate model. In our mediation model, we applied mediation with multiple mediators and multiple independent variables, using the Stata command of ‘sem’ (34).

R2C5:

Why preschool attendance and Year 3 reading/numeracy is stronger among Aboriginals but not non-Aboriginals? Is something in the Aboriginal culture in play here that might explain this result?

We have added the following explanation in our discussion section.

Previous NT study provides “encouraging empirical evidence for increased preschool attendance of Aboriginal children being associated with increased early year school attendance rates and thus better NAPLAN achievement outcomes” (2). The same study also found the greater effect of preschool attendance on early school attendance rates for Aboriginal students than non-Aboriginal students in the NT. The stronger relationship between preschool attendance and Year 3 reading and numeracy for Aboriginal children by comparison with non-Aboriginal children is a function of learning English as a foreign language in many remote communities. The aspirational goal of early childhood system is to achieve equity across all life outcomes and this is reflected in the 2020 Closing the Gap Partnership Agreement. Particular attention is given to how equity in outcomes requires differentiated early childhood programs for Aboriginal and Torres Strait Islander communities. This includes more holistic services, bilingual and culturally inclusive educators. In many community contexts where families may be managing multiple and complex issues, “children’s preschool participation helps parents to build the habit of structuring a typical day around their children’s school routine.”(2) It is possible to design early childhood education provision that recognises the universal benefit for all children, while also taking into account that some children benefit more or require additional support to achieve the same outcome. Known as the proportionate universalism approach, every child would receive a baseline level of preschool provision, and vulnerable children and families would receive extra support. For example, in the 2008 Coalition of Australian Governments’ reform agenda, it was proposed that Aboriginal children would have access to two years of preschool to address a number of areas of need. This did not come to fruition. In the NT, children do have access to publicly provided preschool for a minimal and voluntary parent contribution—"the majority of preschool programs (94%) were delivered free of charge for children aged from 4 years in provincial and remote areas and from 3 years in very remote areas by the NT government.” 

R2C6:

Why do you not report pooled results first (on the entire sample) and then disaggregate by Aboriginal status in subsequent columns? If you have a specific reason then it would be good if you explain it before reporting your results. 

We have revised Table 1 to include descriptive statistics for the whole cohort as well. For all other analyses, we then stratified the results by Aboriginal status due to the different demographic characteristics and living circumstances (i.e. much higher proportion of Aboriginal children living in remote and socio-economic disadvantaged regions. In the Introduction section (considering the unique circumstances in the Northern Territory (NT) of Australia), we have also added a paragraph to explain the unique situation in the NT, in which there are significant differences between the Aboriginal and non-Aboriginal children:

 “The demography of NT Aboriginal children is not only different from their non-Aboriginal peers in the NT, but also different from Aboriginal children in other Australian jurisdictions (35). The majority of Aboriginal children in the NT have language backgrounds other than English and live in remote or very remote regions (i.e. 75-76%), while the majority of non-Aboriginal children in the NT and Aboriginal children in other Australian jurisdictions have English-speaking backgrounds (i.e. 84% and 81% respectively) and do not live in remote or very remote regions (i.e. 76% and 88% respectively) (35, 36). The significant overlap between Aboriginal background and non-English speaking background in the NT (i.e. 75%) is vastly different from other Australian jurisdictions. Brinkman (2012) found that in 2009, less than 1% of all Australian children (excluding the NT) taking Australian Early Development Census (AEDC) assessment (at 2009) had both Aboriginal heritage and non-English speaking backgrounds (36, 37).”

R2C7:

I don't understand the argument behind only having early attendance and early literacy/numeracy as mediators. Also, by “early”, when exactly were these measured or how old were children? At what age self-regulation/executive functions were measured? How were they measured, e.g., did mothers report these? If reported by mothers, should readers be worried about reporting bias?

As stated in our methods, ‘early years attendance’ was defined as the attendance rates from Transition to Year 2 (approximately age 5 to 7/8 years old), and the ‘early literacy/numeracy’ skills was measured at Transition years (approximately age 5). 

In the ‘measure’ subsection in our method section, we have already stated clearly how and when the ‘self-regulation and executive function' was measured: 

“The indicators of self-regulation and executive function at Transition were obtained using items from nine sub-domains in the AEDC”

As such, self-regulation/executive function was not reported by mothers.

We have added the explanation below for the reason of including as mediators in the main text:

“The decision to have early literacy/numeracy and early years attendance as mediators in the pathway was based on two previous studies (2, 38). Collie (2018) found that early literacy/numeracy (at age 5) had a mediating role between prosocial behaviour and Year 3 academic achievement (i.e. NAPLAN) (38), while Silburn (2016) found that early years attendance has a mediating role between preschool attendance and Year 3 NAPLAN for NT Aboriginal children (2).” 

To avoid the confusion to the readers, we have also rewritten our Abstract (methods and result sub-section) to include more information about the age/timing for the different measures in our study.

Methods

This study linked the Australian Early Development Census (AEDC) to the attendance data (i.e. government preschool and primary schools) and Year 3 National Assessment Program for Literacy and Numeracy (NAPLAN). Structural equation modelling was used to investigate the pathway from self-regulation and executive function (SR-EF) at age 5 to early academic achievement (i.e. Year 3 reading/numeracy) at age 8.

Result

The study confirms the expected importance of SR-EF for all children but suggests the different pathways for Aboriginal and non-Aboriginal children. For non-Aboriginal children, there was a significant indirect effect of SR-EF (β=0.38, p<0.001) on early academic achievement, mediated by early literacy/numeracy skills (at age 5). For Aboriginal children, there were significant indirect effects of SR-EF (β=0.19, p<0.001) and preschool attendance (β=0.20, p<0.001), mediated by early literacy/numeracy skills and early primary school attendance (i.e. Transition Years to Year 2 (age 5-7)).

R2C8:

Some important variables are omitted from the model: peer effects (e.g., my Year 3 learning outcomes might have improved due to peers), classroom size/student-teacher ratio (e.g., studies suggest classroom size affects learning outcomes), and “negative” teaching (e.g., frequent punishment and lack of empathy by teachers might also affect learning outcomes). If you have these variables, you could consider using them in the model. If not available, you should consider highlighting these under limitations.

In our study, these variables were unavailable. We have highlighted these limitations in our limitation section.

Finally, the data available to and used in this study did not include other important factors that may influence or modify the outcomes, such as parental involvement in their children’s learning prior to or during preschool, the quality of preschool programs attended or the learning environment (e.g. peer-effects, classroom size, student-teacher ratio or teachers’ teaching style).

R2C9: 

“For non-Aboriginal children, the effect of self-regulation and executive function on Year 3 academic outcomes was mainly mediated by early literacy/numeracy skills. For Aboriginal children, both early years attendance and early literacy/numeracy skills appeared to mediate this effect.” (page 18)

So, this means self-regulation and executive function’s impact on early literacy/numeracy skills are what is affecting Year 3 literacy/numeracy, as self-regulation and executive function have no direct impact. I think what is happening here is that either (i) self-regulation and executive function kick-start the process by impacting immediate learning outcome, and then immediate/previous learning outcomes start affecting future learning outcomes; or, (ii) self-regulation and executive function measured early in life has only an immediate impact on early literacy/numeracy skills, so for Year 3 outcomes, you self-regulation and executive function skills measures among Year 3 children. Can you discard these possibilities? If yes, how?

The well established pathways in the literature about early childhood development lead us to assume that self-regulation and executive function are foundational to early literacy and numeracy skills and the way systems typically measure or prioritise these academic skills may make these outcomes more evident or influential. In our data-linkage study, we are unable to verify this question, and thus unable to discard any of the possibilities the reviewer mentioned. But these can be explored in future studies.

R2C10: 

In Table 2, effect sizes from the “mediation model” do not add up to effect sizes in the “direct path model”. Direct+indirect from mediation should equal to direct from the main model, right? If not, why?

We apologise for the confusion caused. As described in the data-analysis subsection of our Methods section, we used separate models (i.e. direct path model and mediation model). However, we did not report the total effect in the mediation model. We have addressed this concern by reporting the total effect in Table 2.

R2C11:

 You use “National Minimum Standard” to create a dummy for the Year 3 reading/numeracy (page 9), what is the exact cut-off used here? 

We have added more details in the main text.

“Under the NAPLAN assessment scale, there are 10 bands, and the second lowest band reported for each year level represents "the national minimum standard expected of students at that year level", which is "the agreed minimum acceptable standard of knowledge and skills without which a student will have difficulty making sufficient progress at school" (39).

R2C12: 

Do coefficients between Table 3 and Table 4 statistically differ? For example, -0.23 vs -0.19 (male variable, column 1) or -0.37 vs -0.11 (non-English variable, column 2), are these statistically different? 

In our main manuscript, the results (i.e. standardized beta coefficients from the SEM) for both Aboriginal and non-Aboriginal students were presented in Table 3; there is no Table 4. We have conducted our analysis (i.e. involving structural equation model) stratified by Aboriginal status, and not treating Aboriginal status as a variable in the model. Since there were two separate models, we did not conduct statistical significance test for the differences in standardised beta coefficients between Aboriginal and non-Aboriginal children.

To improve the clarify, we have added the following explanation in the ‘analysis’ subsection in our method section.

“Informed by prior research (2, 40) and due to the different demographic characteristics and pathways (to academic outcomes) of Aboriginal and non-Aboriginal children in the NT (as described in the Introduction section), all analyses were stratified by Aboriginal status.” 

 

MINOR COMMENTS:

R2C13: 

It's better to have the main research question in the first or second paragraph of the introduction. Otherwise, the long literature review is quite distracting. 

We understand that the literature review in the introduction could be long, which might make it difficult for readers to follow. To improve the ease of reader for readings, we have added sub-headings in the Introduction and Discussion section. The additional sub-heading, ‘research question of the study’ before the last paragraph of the introduction section should help readers find the research questions of our study.

R2C14:

Without reporting the actual model, saying "...the standardized beta coefficients (β) were reported.." is quite puzzling to the reader. Can you please elaborate on it or, even better, write down the actual model here?

We are using a structural equation model, in which there are two components: structural model (which specifies the predictive relationship among the latent variables) and measurement model (which defines how the latent variables are measure (i.e., represented by indicators)). The standardized beta coefficients (β) are also known as standardised path coefficient. To avoid confusion to the readers, we have simplified the text and added more explanations in the main text.

“In the SEM, the standardised beta coefficients (β), rather than unstandardized beta coefficients, were reported. If the standardised beta coefficient (β) in the pathway from variable A to variable B is 0.5, then for one SD increase in A, B will increase by 0.5 SD. This indicates that when variable A increases by one SD from its mean, variable B can be expected to increase by 0.5 its own SD from its own mean while holding all other relevant variables constant. In reporting unstandardized beta coefficients, when variable A increases by one unit, variable B would be expected to increase by 0.5 unit, while holding all other relevant variables constant. Due to the different scales of the different variables in our study, it is essential to report the standardised beta coefficients to ensure consistent comparison of the path amongst different variables. Standardised beta coefficients (β) equal to or greater than or equal to 0.10 and 0.25 were considered evidence of moderate and large effect size respectively (41, 42).”

 

R2C15:

 You use the abbreviation NAPLAN in methods but don't mention the full form in intro or methods. 

We have mentioned the full form in the first paragraph of our method section.

“Our study cohort consisted of children who had received AEDC assessments (Cycle 1 and Cycle 2 in 2009/10 and 2012 respectively), attended public preschool and school (from first year of formal schooling, the Transition year, to Year 3), and participated in Year 3 National Assessment Program for Literacy and Numeracy (NAPLAN) test in the NT.”

R2C16: 

Various typos and punctuation errors. Please fix those. 

We have fixed the various typos and punctuation errors in the manuscript.

Reviewer 3

R3C1: 

The topic being studied is of great importance. Understanding better the precise ways in which investments in early childhood can improve student skills is vital to the design of better public policy investments in this area. Within this broad umbrella, exploring the precise specific constraints faced by children from marginalized communities is extremely important.

We thank Reviewer 3 for the positive feedback.

R3C2:

 I also appreciate the care the authors have taken to use well-established administrative data and bring different pieces of the data together to create a large representative sample

We thank Reviewer 3 for the positive feedback.

R3C3: 

One point of confusion is the relationship between three different factors: (i) preschool attendance; (ii) self-regulation and executive function; and (iii) early academic achievement (literacy/numeracy). As I understand it, the results seem to suggest that pre-school attendance mediates the relationship between (ii) and (iii) for Aboriginal children but not so much for non-Aboriginal children. I would have really liked to see a clear exploration of the impact of pre-school attendance on self-regulation and executive function – and how this varies between Aboriginal and non-Aboriginal children. I am not sure this aspect comes out clearly in the paper. Once we are clearer on this relationship, I believe we can interpret the broader results on how the pre-school attendance is mediating the relationship between self-regulation and executive function and early academic achievement – and how it varies between Aboriginal and non-Aboriginal children. More fundamentally, I think the authors should map out a clear theory of change between the main relationships modelled.

As stated in our method section previously, early years attendance and early literacy/numeracy skills are mediators (“In the mediation model, we investigated the mediation effects of early literacy/numeracy skills and early years attendance”). As such, pre-school attendance is not a mediator. We have added the following explanation to improve the clarity.

“The decision to have early literacy/numeracy and early years attendance as mediators in the pathway was based on two previous studies (2, 38). Collie (2018) found that early literacy/numeracy (at age 5) had a mediating role between prosocial behaviour and Year 3 academic achievement (i.e. NAPLAN) (38), while Silburn (2016) found that early years attendance has a mediating role between preschool attendance and Year 3 NAPLAN for NT Aboriginal children (2).” 

R3C4: 

Another thing that bothered me was that we don’t see the extent to which the likelihood of being remote and non-English speaking varies between Aboriginal and Non-Aboriginal children.

We understand that international audience might not be familiar to the unique circumstances of the NT, in which most Aboriginal people in the NT lived in remote areas and have non-English speaking background, and the proportion of non-Aboriginal people living in remote areas is higher when compared with their counterparts living in other parts of Australia and internationally. As such, we have added an additional paragraph in the introduction (Unique circumstances in the Northern Territory (NT) of Australia) to provide more NT’s context.

“The demography of NT Aboriginal children is not only different from their non-Aboriginal peers in the NT, but also different from Aboriginal children in other Australian jurisdictions (35). The majority of Aboriginal children in the NT have language backgrounds other than English and live in remote or very remote regions (i.e. 75-76%), while the majority of non-Aboriginal children in the NT and Aboriginal children in other Australian jurisdictions have English-speaking backgrounds (i.e. 84% and 81% respectively) and do not live in remote or very remote regions (i.e. 76% and 88% respectively) (35, 36). The significant overlap between Aboriginal background and non-English speaking background in the NT (i.e. 75%) is vastly different from other Australian jurisdictions. Brinkman (2012) found that in 2009, less than 1% of all Australian children (excluding the NT) taking Australian Early Development Census (AEDC) assessment (at 2009) had both Aboriginal heritage and non-English speaking backgrounds (36, 37).”

In addition, we have rewritten the Background sub-section in our Abstract to provide more NT’s context.

“With the pending implementation of the Closing the Gap 2020 recommendations, there is an urgent need to better understand the contributing factors of, and pathways to positive educational outcomes for both Aboriginal and non-Aboriginal children. This deeper understanding is particularly important in the Northern Territory (NT) of Australia, in which the majority of Aboriginal children lived in remote communities and have language backgrounds other than English (i.e. 75%).”

R3C5: 

Having said that, some of the key policy recommendations seem sound. I just think the authors could do more to persuade the reader that the way the relationships are being examined makes sense.

We have restructured the discussion in which we present the major policy and program implications, by adding additional paragraphs. Please refer to the added paragraphs in our response to Comment 4 of Reviewer 1. In addition, we have re-written the conclusion of the abstract and main text to reiterate the implications of our findings.

“With the implementation of the Closing the Gap 2020 recommendations, there is an urgent need to better understand self-regulation and executive functions as contributing factors to positive educational outcomes for children living in both urban and remote settings. This study had access to linked data of preschool attendance, AEDC, early years attendance and NAPLAN scores, and so was able to provide a basic understanding of the pathways to early academic achievements for both Aboriginal and non-Aboriginal children in the NT. This study acknowledges that NAPLAN is a narrow criterion for school success. Due to data limitations, this study does not provide insights into the pathways to other important positive schooling outcomes (e.g. well-being, aspirations, participation, identities, relational). Currently in Australia, only AEDC, NAPLAN, school enrolment and attendance data are collected nationally in the early childhood and primary education setting. The current study forms the basis for further investigation into self-regulation and executive function as contributing factors to positive educational outcomes for both Aboriginal and non-Aboriginal children in the NT and across Australia. It suggests the need for more attention to self-regulation and executive function in national data-collection.

Despite the limitations, our study offers valuable insights to better understand the contribution of early foundational skills that comprise self-regulation and executive function to positive educational outcomes in different populations. The results demand further investigation to culturally, linguistically and contextually differentiated programs and policies in the current Australian education context. Our study confirms the expected importance of self-regulation and executive functioning skills for all children but suggests there are different pathways for Aboriginal and non-Aboriginal children in the NT. Our study suggested the importance of preschool and early years attendance in the pathway to academic achievement, particularly for Aboriginal children. Further, these results reflect the distinct population profile of the NT with a majority of Aboriginal children with language backgrounds other than English, living in geographically remote communities (i.e. 75%) and with substantial disadvantaged subgroups of children from rural and remote backgrounds in the major centres who have poor access to services, different from other Australian and international jurisdictions. There are potentially cultural or linguistic assets and strengths that contribute to self-regulation and executive function as foundational skills for academic learning that are not recognised in the current tools. 

The complex inter-relatedness of school attendance, remoteness, non-English speaking background and socio-economic status on the pathway for self-regulation and executive function skills demand attention in the design of effective policies and programs. Policy makers and educators must recognise that the factors contributing to non-attendance are complex, hence the solutions require multi-sectoral collaboration in place-based design for effective implementation, particularly for early childhood experiences. Given the importance of self-regulation and executive function for foundational skills, and readiness for academic engagement, there is a pressing need to better understand how current policies and programs enhance children and their families’ sense of safety and support to nurture these skills.”

 

Reference

1. Brinkman SA, Gregory TA, Goldfeld S, Lynch JW, Hardy M. Data resource profile: the Australian early development index (AEDI). International journal of epidemiology. 2014;43(4):1089-96.

2. Silburn S, Guthridge S, McKenzie J, Su J-Y, He V, Haste S: Early Pathways to School Learning: Lessons from the NT Data-Linkage Study: Darwin: Menzies School of Health Research. 2018. Available at: https://www.menzies.edu.au/icms_docs/293933_Early_Pathways_to_School_Learning_%E2%80%93_Lessons_from_the_NT_data_linkage_study.pdf.

3. Foley, M., Zhao, Y., & Condon, J. (2012). Demographic data quality assessment for Northern Territory public hospitals, 2011: Health gains planning, Dept. of Health. Darwin.

4. Australian Institute of Health and Welfare and Australian Bureau of Statistics 2012. National best practice guidelines for data linkage activities relating to Aboriginal and Torres Strait Islander people. AIHW Cat. No. IHW 74. Canberra: AIHW. Available at: https://www.aihw.gov.au/getmedia/6d6b9365-9cc7-41ee-873f-13e69e038337/13627.pdf.

5. Department of Education. Education Engagement Strategy. 2021. Available at: https://education.nt.gov.au/statistics-research-and-strategies/education-engagement-strategy.

6. Prout Quicke S, Biddle N. School (non-) attendance and ‘mobile cultures’: theoretical and empirical insights from Indigenous Australia. Race Ethnicity and Education. 2017;20(1):57-71.

7. Shonkoff, J. P., & Phillips, D. A. (2000). From neurons to neighbourhoods: The science of early childhood development. Washington DC: National Academy Press. .

8. Sabates, R. and Yardeni, A. Chapter 2 Social determinants of health and education: Understanding the intersectionalities during childhood, in R. Midford et al. (eds.), Health and Education Interdependence. Singapore: Springer Nature. 2020.

9. Silburn, S. Chapter 16 The role of epigenetics in shaping the foundations of children's learning, in R. Midford et al. (eds.), Health and Education Interdependence. Singapore: Springer Nature. 2020.

10. Cahill, R. and Dadvand, B. Chapter 11 Social and emotional learning and resilience education, in R. Midford et al. (eds.), Health and Education Interdependence. Singapore: Springer Nature. 2020.

11. Department of Education. NT Social and Emotional Learning. 2021. Available from: https://education.nt.gov.au/support-for-teachers/nt-social-and-emotional-learning.

12. Productivity Commission. Expenditure on Children in the Northern Territory, Study Report, Canberra. 2020. Available at: https://www.pc.gov.au/inquiries/completed/nt-children/report.

13. Wise S. Improving the early life outcomes of Indigenous children: implementing early childhood development at the local level Closing Gap Clear. Canberra: Australian Institute of Health and Welfare/Australian Institute of Family Studies. 2013.

14. Northern Territory Department of Education (NTDE) (1999). Learning lessons: An independent review of Indigenous education in the Northern Territory. Darwin, NT: Northern Territory Department of Education. Available at: https://www.voced.edu.au/content/ngv:11688 at 28 Apr 2021.

15. Ball J. Indigenous Early Childhood development programs as “hook” and “hub” for inter-sectoral service delivery. Variegations: New Research Directions in Human and Social Development. 2003;1:3-9.

16. Cleary, V. Education and Learning in an Aboriginal Community. Issue Analysis, 65, 1-16. 2005.

17. Walker, K. National Preschool Education Enquiry Report. For All Our Children. Melbourne: Australian Education Union. 2004.

18. SNAICC. Research Priorities for Indigenous Children and Youth. Fitzroy: Secretariat of National Aboriginal and Islander Child Care. 2004.

19. Aslam, H., & Kemp, L. Home visiting in South Western Sydney: An integrative literature review, description and development of a generic model. Sydney: Centre for Health Equity Training Research and Evaluation. 2005.

20. Brady, W. (1991). The Health of Young Aborigines: A report on the health of Aborigines aged 12 to 25 years. Canberra: Australian Institute of Aboriginal and Torres Strait Islander Studies.

21. Penman, R. The 'growing up' of Aboriginal and Torres Strait Islander children: a literature review. Canberra: Australian Government. 2006.

22. Hetzel, D., Page, A., Glover, J., & Tennant, S. Inequality in South Australia, Key Determinants of Wellbeing, Volume 1, The Evidence. Adelaide: Department of Health. 2004.

23. Arnold, C., Bartlett, K., Gowani, S., & Merali, R. Is everybody ready? Readiness, transition and continuity: Reﬂections and moving forward (Working Paper 41). The Hague, NL: Bernard van Leer Foundation. 2007.

24. Bamblett, M., Bath, H., & Roseby, R. Growing Them Strong, Together: Promoting the safety and wellbeing of the Northern Territory’s children, Summary Report of the Board of Inquiry into the Child Protection System in the Northern Territory 2010. Darwin, NT: Northern Territory Government. 2010.

25. Ball J, Pence A, Benner A. Quality child care and community development: What is the connection. Too small to see, too big to ignore: Child health and well-being in British Columbia. 2002;35:75-102.

26. Edwards, B., Wise, S., Gray, M., Hayes, A., Katz, I., Misson, S., ... Muir, K. Stronger Families in Australia Study: The Impact of Communities for Children (Occasional Paper 25) Canberra: Department of Families, Housing, Community Services and Indigenous Affairs. 2009.

27. McRae, D., Ainsworth, G., Cumming, J., Hughes, P., Mackay, T., Price, K., ...Zbar, V. What Works: Explorations in improving outcomes for Indigenous students. Canberra: Australian Curriculum Studies Association. 2000.

28. Fasoli, L., Benbow, R., Deveraux, K., Falk, I., Harris, R., Hazard, M., ... Railton, K. ‘Both Ways’ Children’s Services Project. Batchelor, NT: Batchelor Institute of Indigenous Tertiary Education. 2004.

29. Centre for Community Child Health. Integrating services for young children and their families. Policy Brief, 17. Parkville, Victoria: Royal Children’s Hospital. 2009.

30. Allison PD, editor Handling missing data by maximum likelihood. SAS global forum; 2012.

31. Henseler J. Composite-based structural equation modeling: analyzing latent and emergent variables: Guilford Publications; 2020.

32. Petter S, Straub D, Rai A. Specifying formative constructs in information systems research. MIS quarterly. 2007:623-56.

33. Fassott G, Henseler J, “Formative (Measurement),” in Wiley Encyclopedia of Management, Vol. 9, Marketing, Cary Cooper, Nick Lee, and Andrew Farrell, eds., Chichester: Wiley, 1–4. 2015.

34. UCLA Institute for Digital Research & Education. How can I do mediation analysis with the SEM command? | Stata FAQ. Available at: https://stats.idre.ucla.edu/stata/faq/how-can-i-do-mediation-analysis-with-the-sem-command/.

35. Australian Bureau of Statistics. 3238.0.55.001-Estimates of Aboriginal and Torres Strait Islander Australians, June 2016. 2018 Available from: https://www.abs.gov.au/statistics/people/aboriginal-and-torres-strait-islander-peoples/estimates-aboriginal-and-torres-strait-islander-australians/latest-release.

36. Centre for Community Child Health and Telethon Institute for Child Health Research. A Snapshot of Early Childhood Development in Australia – AEDI National Report 2009, Australian Government, Canberra. 2009. Available at: https://www.aedc.gov.au/resources/detail/national-report-2009.

37. Brinkman SA, Gialamas A, Rahman A, Mittinty MN, Gregory TA, Silburn S, et al. Jurisdictional, socioeconomic and gender inequalities in child health and development: analysis of a national census of 5-year-olds in Australia. BMJ open. 2012;2(5):e001075.

38. Collie RJ, Martin AJ, Roberts CL, Nassar N. The roles of anxious and prosocial behavior in early academic performance: A population-based study examining unique and moderated effects. Learning and Individual Differences. 2018;62:141-52.

39. Australian Curriculum Assessment and Reporting Authority 2018, NAPLAN Achievement in Reading, Writing, Language Conventions and Numeracy: National Report for 2018, ACARA, Sydney.

40. Robinson, G; William, T. Chapter 8 The Child, Between School, Family and Community: Understanding the Transition to School for Aboriginal Children in Australia’s Northern Territory, in R. Midford et al. (eds.), Health and Education Interdependence. Singapore: Springer Nature. 2020.

41. Keith T. Structural equation modeling in school psychology. The handbook of school psychology. 1999;3(1):78-107.

42. Keith TZ. Multiple regression and beyond: An introduction to multiple regression and structural equation modeling: Routledge; 2014.

---

## [Decision Letter · Decision Letter 1]

28 Oct 2021

Pathways to school success: Self-regulation and executive function, preschool attendance and early academic achievement of Aboriginal and non-Aboriginal children in Australia’s Northern Territory

PONE-D-21-15913R1

Dear Dr. He,

We’re pleased to inform you that your manuscript has been judged scientifically suitable for publication and will be formally accepted for publication once it meets all outstanding technical requirements.

Kind regards,

Nishith Prakash, Ph.D.

Academic Editor

PLOS ONE

Additional Editor Comments (optional):

Dear Dr. He,

Delighted to accept this paper. I think this work of yours makes valuable contribution and I am glad you chose PLOS ONE.

Best,

Nishith

Reviewers' comments:

Reviewer's Responses to Questions

**Comments to the Author**

1. If the authors have adequately addressed your comments raised in a previous round of review and you feel that this manuscript is now acceptable for publication, you may indicate that here to bypass the “Comments to the Author” section, enter your conflict of interest statement in the “Confidential to Editor” section, and submit your "Accept" recommendation.

Reviewer #1: All comments have been addressed

Reviewer #2: All comments have been addressed

2. Is the manuscript technically sound, and do the data support the conclusions?

Reviewer #1: Yes

Reviewer #2: Yes

3. Has the statistical analysis been performed appropriately and rigorously? 

Reviewer #1: Yes

Reviewer #2: Yes

4. Have the authors made all data underlying the findings in their manuscript fully available?

Reviewer #1: Yes

Reviewer #2: (No Response)

5. Is the manuscript presented in an intelligible fashion and written in standard English?

Reviewer #1: Yes

Reviewer #2: Yes

6. Review Comments to the Author

Reviewer #1: (No Response)

Reviewer #2: (No Response)

7. PLOS authors have the option to publish the peer review history of their article (what does this mean?). If published, this will include your full peer review and any attached files.

Reviewer #1: No

Reviewer #2: No

---

## [Editor Report · Acceptance letter]

2 Nov 2021

PONE-D-21-15913R1 

Pathways to school success: Self-regulation and executive function, preschool attendance and early academic achievement of Aboriginal and non-Aboriginal children in Australia’s Northern Territory 

Dear Dr. He:

I'm pleased to inform you that your manuscript has been deemed suitable for publication in PLOS ONE. Congratulations! Your manuscript is now with our production department. 

Kind regards, 

on behalf of

Dr. Nishith Prakash 

Academic Editor

PLOS ONE